# An Efficient One-Class SVM for Novelty Detection in IoT

**Kun Yang** [*]                                                               *ky2440@columbia.edu*
*Columbia University*

**Samory Kpotufe**                                                             *skk2175@columbia.edu*
*Columbia University*

**Nick Feamster**                                                             *feamster@uchicago.edu*
*University of Chicago*

**Reviewed on OpenReview:** *https://openreview.net/forum?id=LFkRUCalFt*

## Abstract

One-Class Support Vector Machines (OCSVMs) are a set of common approaches for novelty detection due to their flexibility in fitting complex nonlinear boundaries between normal and novel data. Novelty detection is important in the Internet of Things ("IoT") due to the potential threats that IoT devices can present, and OCSVMs often perform well in these environments due to the variety of devices, traffic patterns, and anomalies that IoT devices present. Unfortunately, conventional OCSVMs can introduce prohibitive memory and computational overhead in detection. This work designs, implements, and evaluates an efficient OCSVM for such practical settings. We extend Nyström and (Gaussian) Sketching approaches to OCSVM, combining these methods with clustering and Gaussian mixture models to achieve 15-30x speedup in prediction time and 30-40x reduction in memory requirements without sacrificing detection accuracy. Here, the very nature of IoT devices is crucial: they tend to admit few modes of *normal* operation, allowing for efficient pattern compression.

## 1 Introduction

As devices ranging from consumer electronics to building control systems become connected to the Internet as part of the "Internet of Things" (IoT), both these devices and the network itself are subject to new threats. Novelty detection, which aims to detect unusual activity based on observable properties of network traffic, is a common defense. One-Class Support Vector Machines (OCSVMs) are one of the common approaches for novelty detection[1] due to their ability to identify a wide range of nonlinear classification boundaries. Such flexibility is appropriate for IoT devices and applications, which exhibit complexity due to the vast heterogeneity of devices and the wide range of traffic patterns under different operating modalities.

The heterogeneity of IoT devices and operating regimes introduces a broad class of activities (and corresponding network traffic patterns) that could be classified as normal or novel. In contrast to general-purpose computing devices—where the main novel behavior of interest is typically a security event such as an infection—IoT devices raise a more diverse set of anomalies, including physical device failures, the introduction of rogue devices on the network, physical security incidents, and abnormal interactions with control systems. The devices themselves are also heterogeneous, with the normal operating regime for each device type or manufacturer exhibiting distinct normal baseline patterns. Given such diversity of anomaly patterns, desired anomaly

---

[*]The work was done at Columbia University.

[1]In the context of security, novelty detection is often referred to as *anomaly detection*; we use the term *novelty detection* to refer to the same class of algorithms, as the problem is equivalent. We prefer the use of novelty detection in this paper because the classes of events that we aim to detect include conventional anomalies (in the security sense), as well as a broader class of novel events, e.g., activities, and devices that might be simply "new", though these new events may not necessarily have a negative connotation.

detection methods ought to be flexible in the generality of anomalies they can fit, but easy to use. OCSVM is a clear candidate meeting these criteria and has thus been frequently applied to novelty detection problems in IoT, with demonstrable efficacy in detecting novel traffic patterns corresponding to either unseen modalities or malicious activities (Shilton et al., 2015; Lee et al., 2016; Mahdavinejad et al., 2018; Al Shorman et al., 2020; Razzak et al., 2020). We illustrate OCSVM's detection performance against some other flexible methods on typical IoT devices in Figure 1, with further discussion in (Related Work) Section 2.

However, many IoT deployments require *fast* novelty detection in field deployments (e.g., embedded devices such as home network routers or embedded sensors), where both computational and memory requirements may be limited. In operational deployments, there may be the need to quickly detect an attack, a rogue device, or a novel activity (perhaps non-malicious). Unfortunately, OCSVMs can be computationally expensive at *detection time*. Given a new observation $x$ to classify as normal or novel, detection consists of evaluating a scoring function $f(x)$—of the form $\sum_{i=1}^{n} \alpha_i K(X_i, x)$, defined with respect to training data $\{X_i\}$ of size $n$ and a so-called *kernel function $K$*; such evaluation of $f(x)$ takes time and space $\Omega(n)$ for typically large training data size $n$ in the thousands. In the context of IoT, each training data point $X_i$ represents a vectorized representation of *normal* traffic data over short time periods. Given an Internet-connected device that is continuously generating network traffic, detection using OCSVM is currently prohibitive in practice.

**Goal and Method.** The goal of this work is to speed up detection time and reduce memory requirements of OCSVM, *while maintaining detection performance*; we demonstrate these gains in the context of IoT, which imposes time and memory constraints in practice. Our focus is on *detection* time and space, rather than *training* time and space, and we particularly emphasize that *we seek to not tradeoff detection performance*, in contrast to usual expectations for computational speedup.

Although novelty detection is an *unsupervised problem*—i.e., we only have access to *normal data* as opposed to both normal and novel data points—we draw initial inspiration from the related *supervised learning* methods of Support-Vector-Machines (SVMs), which, similarly to OCSVMs, uncovers linear relationships between classes of data. Namely, various speedup approaches such as so-called Nyström and Sketching (Drineas et al., 2005; Yang et al., 2017) have recently been developed for SVMs, which we aim to build on. **However, we will argue that such speedup approaches cannot be applied as usual in the unsupervised case considered here:** in particular, while it is usually expected that a linear decision boundary is fit after the Nyström or Sketching (compression) step, we argue that we need to instead fit nonlinear *clustering* boundaries in the unsupervised regimes considered here, *if we are to maintain detection accuracy.*

To better understand relevant distinctions between unsupervised and supervised OCSVMs in the context of speedup methods, we need to get into a bit more detail. Most significantly, these methods all operate on a so-called *gram matrix $\mathcal{K} \in \mathbb{R}^{n \times n}$*, encoding relations between data points, i.e., inner-products $\mathcal{K}_{i,j} \doteq \phi(X_i) \cdot \phi(X_j)$ corresponding to an implicit data transformation $x \mapsto \phi(x)$. Operations on $\mathcal{K}$ are often the bottleneck in training and prediction time, and approaches such as *Nyström* and *Sketching* approximate $\mathcal{K}$ with a lower-rank matrix $\mathcal{K}'$ that allows faster operations, while nearly preserving the original relations between transformed data points $\phi(X_i)$'s. In particular, in the case of SVM, the Nyström or Sketching matrix $\mathcal{K}'$ results in a new transformation $x \mapsto \phi(x) \mapsto \phi'(x)$ where for any two data points $X_i, X_j$ from separate classes, $\phi'(X_i)$ and $\phi'(X_j)$ remain *linearly separated* if $\phi(X_i)$ and $\phi(X_j)$ were linearly separable. In other words, one can simply proceed similarly with $\mathcal{K}'$ in place of $\mathcal{K}$ and train a linear classifier. Unfortunately, as we will see (Section 3.2), in the unsupervised case of OCSVM, such $\phi'(X_i)$ and $\phi'(X_j)$ might no longer be linearly separable—an issue particularly true in IoT—requiring us to develop a different approach on top of Nyström or Sketching.

To address this issue of nonlinearity in the case of OCSVM after applying Nyström or Sketching, we rely on recent interpretations of these speedup approaches (Yang et al., 2012; Rudi et al., 2015; Kpotufe & Sriperumbudur, 2020) which show that the resulting transformation $\phi'$ preserves *distances* between transformed points $\phi(X_i), \phi(X_j)$ (i.e., distances in the transformed kernel space $\{\phi(x)\}$), even if linear separability between classes is not preserved. As such $\phi$-distances are preserved, one might then expect that *cluster* structures that may be apparent under $\phi$-transformation remain apparent under $\phi'$. Building on this intuition, detection will therefore just consist of flagging any future query point $x$ as *abnormal* if $\phi'(x)$ falls far from clusters in the remapped training data $\{\phi'(X_i)\}_{i=1}^{n}$. To implement this idea, we model clusters in $\{\phi'(X_i)\}_{i=1}^{n}$ as components of a Gaussian Mixture Model (GMM), which has the benefit of allowing for

a simple detection rule based on *density levels* (see Section 4). Finally, as the GMM model introduces a new hyperparameter on top of vanilla OCSVM, namely, the number $k$ of Gaussian components (or number of clusters), we further propose a basic approach to automatically set the parameter $k$ by estimating high density regions of $\{\phi'(X_i)\}_{i=1}^{n}$ via existing methods such as *QuickShift++* (Jiang et al., 2018).

**Results Overview.** We implement the above described approach based on mapping the *normal* training data as $\{\phi'(X_i)\}_{i=1}^{n}$ using either Nyström or a simple form of Sketching termed *Kernel Johnson-Linderstrauss* (KJL) shown recently to preserve cluster structures w.r.t. the original mapping $\phi$ induced by kernel methods such as OCSVM (Kpotufe & Sriperumbudur, 2020). For simplicity, we will henceforth refer to these approaches respectively as OC-Nyström and OC-KJL, where *OC* stands for *One Class* (as in OCSVM) to emphasize the unsupervised nature of these methods. We evaluate OC-Nyström and OC-KJL, both with and without automatic GMM parameter selection, on multiple IoT datasets encoding a variety of detection use cases of interest from detecting benign new devices to malicious activity from infected devices.

To evaluate the effectiveness of our techniques in the context of IoT anomaly detection, we test against both public network datasets that apply to IoT environments and datasets that we have generated in the lab based on common interactions with consumer IoT devices. In addition to IoT-specific datasets, we also evaluate our algorithms on several public datasets involving traffic generated by general-purpose computing devices that are pertinent to IoT settings, including distributed denial of service (DDoS) attack detection and novel device activity. *The very nature of IoT device's network behavior facilitates faster detection time and space:* typical IoT devices, e.g., smart appliances and traffic monitors, have few operational modalities, inducing few *clusters* of normal traffic; as a result, we can expect a small number $k$ of clusters, i.e., GMM components needed to faithfully model normal operational traffic, leading to a smaller memory footprint and detection time complexity. Our results are as follows:

- *Significant reduction in detection time and space.* We observe typical detection time speedups (w.r.t. the baseline OCSVM) between 14 to 20 times faster using either OC-Nyström or OC-KJL, and 40+ times for some datasets. Typical space complexities decrease by a factor of 20 or more w.r.t. OCSVM. In particular, for computationally-constrained deployment platforms, e.g., a home router, OCSVM detection time[2] is typically in the order of 100 ms per data point (see Table 3), which is now reduced to about 5 ms per data point; combined with the reduction in space, this opens up processing multiple data points from multiple IoT devices simultaneously on the same router, while maintaining detection performance, as explained in the next bullet point.

- *Equivalent or improved detection performance.* Given that detection performance of any machine learning method depends on hyperparameter choices, we consider two situations: (1) a situation where hyperparameters are adequately calibrated using side data (i.e., a small validation set independent of future test data), and (2) a situation where such side data might be missing and basic rules-of-thumb are employed to select hyperparameters. Such a situation might arise in IoT settings where some activities and devices might be labeled, but the vast majority remain unlabeled due to the dataset scale and heterogeneity.

  Upon proper calibration of all three procedures, both OC-Nyström or OC-KJL achieve detection performance on par with the baseline OCSVM as measured by area under the curve (AUC). Both slightly outperform OCSVM in some cases, which is likely due to the fact that the new mapping $\phi'$ acts as a lower-dimensional projection which at times recovers intrinsic structure not present in abnormal traffic.

  In the second situation, i.e., under rule-of-thumb choices of the main hyperparameter shared by all three procedures (i.e., a so-called *kernel bandwidth* parameter), OC-Nyström and OC-KJL (with automatic choices of number of GMM components $k$) attain at least 0.85% of OCSVM's AUC on most datasets and manage improvements in AUC over that of OCSVM on many datasets. Given the lack of proper calibration, we observe some rare situations where AUC degrades more considerably compared to OCSVM. We include these results to give a fair and broad sense of the range of performance that one could potentially observe in practice.

**Paper Outline.** The rest of the paper is organized as follows. In Section 2 we go over related work on novelty detection in computer networking and for IoT in particular, and further discuss the appeal of OCSVM

---

[2]Such single-board computers have similar specifications as Raspberry PI or Nano devices used in our experiments.

in this domain. Section 3 gives more detailed background on OCSVM, and relevant intuition on Nyström and KJL; we then build on this intuition in Section 4 to derive OC-Nyström and OC-KJL approaches in detail. In Section 5 we describe our experimental setup, including preprocessing choices for encoding network traffic as vectors, and evaluation metrics in detail. This is followed by experimental results in Section 6.

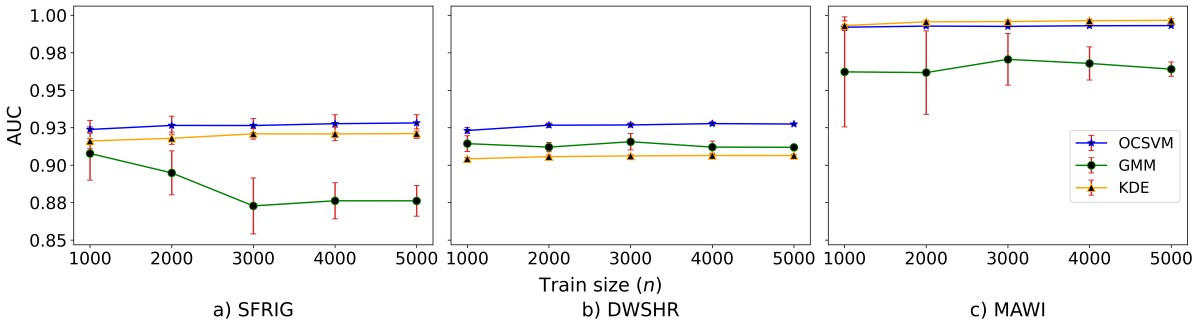

**Figure 1:** *Detection performance of OCSVM vs. that of GMM and KDE on two IoT datasets (a and b) and on general-purpose PC (c). OCSVM remains competitive against these other classical and flexible methods (GMM and KDE). Shown on all datasets, are the best performances obtained across hyperparameter choices (such choices are described in Section 5.1).*

## 2 Related Work

We survey related work in anomaly detection in networking, both generally and in the context of IoT.

**Network Anomaly Detection.** Anomaly detection in networks is widely studied. Ahmed et al. provide a more complete survey of these techniques (Ahmed et al., 2016); we briefly overview some of the general classes of techniques. Various supervised learning techniques have been applied to the problem of network anomaly detection, including Support Vector Machines (Eskin et al., 2002), Bayesian networks (Kruegel & Vigna, 2003), sequential hypothesis testing (Jung et al., 2004), and neural networks (Hawkins et al., 2002; Wang et al., 2017). In many of these cases, supervised learning has been applied in a very specific context when labeled datasets were available, such as detecting port scans (Jung et al., 2004) or web-based attacks (Kruegel & Vigna, 2003). OCSVM remains a common technique for performing anomaly detection in IoT, having been used in a variety of contexts, including sensor networks Rajasegarar et al. (2010), intrusion detection of system calls Heller et al. (2003), network intrusion detection Zhang et al. (2015), and anomaly detection in wireless sensor networks Zhang et al. (2009).

Common unsupervised approaches have involved principal component analysis (Shyu et al., 2003; Lakhina et al., 2004) and generalized likelihood ratio (Thottan & Ji, 2003). Principal component analysis in particular has proved problematic in the context of network anomaly detection due to the fact that transforming network traffic into a matrix representing a multidimensional time series involves quantization and discretization that render the resulting underlying models brittle (Ringberg et al., 2007). Ringberg et al. found that when applying PCA to network traffic anomaly detection, the false positive rate is sensitive to the selection of the number of principal components in the normal subspace and the level of traffic aggregation (Ringberg et al., 2007).

**Anomaly Detection in IoT.** Unsupervised learning approaches are a popular approach for IoT, where obtaining detailed labels for a large, heterogenous set of devices is impractical. Over the past several years, unsupervised learning techniques have been developed for novelty detection specifically for IoT devices and activities; One-class SVMs (OCSVMs) have been particularly effective for detecting anomalies in IoT settings (Shilton et al., 2015; Lee et al., 2016; Mahdavinejad et al., 2018; Al Shorman et al., 2020; Razzak et al., 2020). OCSVMs are appropriate for novelty detection in IoT due to their ability to learn complex, nonlinear decision boundaries, which can be important in IoT environments where activities are diverse and heterogeneous. Note that OCSVMs do not always have superior performance, as alternative methods

can indeed be better on given IoT device's datasets; however, they've received much attention in the IoT domain as they are simple to operate and compete well across a variety of IoT problems as argued in the aforementioned works. Such flexibility and competitiveness is illustrated in Figure 1 vs. other classical (flexible) novelty detection approaches.

Unfortunately, despite its efficacy in these settings, OCSVM can be particularly costly in terms of both time and memory requirements, rendering the previous work impractical for many deployment settings where novelty detection algorithms would be deployed in practice. Specifically, IoT deployments involve the deployment of resource-constrained devices; in the case of consumer IoT deployments, for example, anomaly detection systems may need to operate on home routers, where processing and memory capacity is limited. The algorithms we develop in this paper achieve a speedup of up to 40 times as compared to the best-known implementations of OCSVM, thus making it possible to deploy these anomaly detection algorithms in practice in IoT settings. To demonstrate this feasibility, we evaluate the real-time performance and memory requirements of our algorithms on embedded single-board computers that are often deployed in home network settings.

## 3  Background on Methods

### 3.1  (Gaussian Kernel) OCSVM

**Basic Background.** OCSVM first maps data $x \in \mathbb{R}^D$ as $\phi(x)$ into an infinite dimensional space $\mathcal{H}$ (a so-called *reproducing kernel Hilbert space* (RKHS)). As a Hilbert space, $\mathcal{H}$ admits basic vector operations as in Euclidean $\mathbb{R}^D$, in that it has a well-defined inner-product $\langle \phi(x), \phi(x') \rangle$ inducing a norm $\|\phi(x)\|^2 = \langle \phi(x), \phi(x) \rangle$ and hence a notion of distance between points and space geometry (clusters, linear projections, hyperplanes, spheres, etc). All that is therefore needed for geometric operations is access to the inner-product operation $\langle \cdot, \cdot \rangle$, which is readily provided by RKHS theory: for any data points $x, x' \in \mathbb{R}^D$, there exists a so-called *kernel function* $K$ satisfying $K(x, x') = \langle \phi(x), \phi(x') \rangle$. Therefore, given access to $K$, the mapping $\phi$ need not be explicitly computed, as all geometric operations are implicit through $K$ alone, and in particular, all geometric operations involved in learning a hyperplane separating classes of points are thus determined by $K$ alone. The most common kernel function in machine learning, and especially in OCSVM, is the *Gaussian* kernel $K(x, x') = C \cdot \exp\left(-\|x - x'\|^2 / 2h^2\right)$ (for a *bandwidth* hyperparameter $h$ to be chosen in practice, and a normalizing constant $C = C(h)$).

**Key Intuition and Operations.** A main intuition behind the mapping $\phi$, implicit in both supervised SVM and unsupervised OCSVM, is that it manages to *separate* classes of data, i.e., pull corresponding data points far apart in $\mathcal{H}$, even when they are not easily separable in their original representation in $\mathbb{R}^D$. This is illustrated in Figure 2.

It follows that, after the mapping $\phi$, the data might become linearly separable in $\mathcal{H}$, i.e., the two classes of data, *normal* and *abnormal*, fall on different sides of a hyperplane in $\mathcal{H}$. Therefore, in supervised learning (e.g., with SVM) where we have access to both classes of data at training time, we simply would learn a hyperplane that most faithfully separates the training data into the two class labels. However, in the case of OCSVM, only one class is available during training, namely, *normal* data. It is therefore unclear how to separate it from unseen *anomaly* data. The

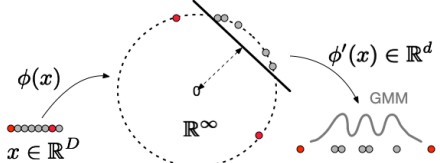

**Figure 2:** *OCSVM maps data points $x \in \mathbb{R}^D$ as $\phi(x)$ onto a ball in $\mathbb{R}^\infty$, inducing linear separation between normal points (gray) and yet unseen novel points (in red). We consider mappings $\phi'$, which then remap down to $\mathbb{R}^d$, $d \ll D$, while maintaining separation (into clusters), but not necessarily linear separability. Given the few modalities of IoT devices, we can then learn a GMM with few components to model the remapped* normal *data.*

main insight is that, if the kernel $K$ satisfies $K(x, x) = C$ for some constant $C$, as with the Gaussian kernel, then all points $x \in \mathbb{R}^D$ are mapped in $\mathcal{H}$ to the surface of a sphere of radius $\sqrt{C}$, since $K(x, x) = \|\phi(x)\|^2 = C$. It follows that if the two classes are linearly separable, then they can be separated by a hyperplane that puts

a maximal margin between the *normal* class and the center of the sphere since unseen anomaly data is also constrained to map to the surface of the sphere. This is illustrated in Figure 2.

OCSVM, thus, using normal data $\{X_i\}_{i=1}^{n}$ alone, returns a hyperplane that isolates normal data from future anomalous observations. Such a hyperplane can be estimated without actually computing $\phi(X_i) \in \mathcal{H}$, simply through geometrical operations encoded by all pairwise inner-products $\langle \phi(X_i), \phi(X_j) \rangle$ given by $K(X_i, X_j)$. These inner products are encoded for convenience in a so-called *gram matrix* $\mathcal{K} \in \mathbb{R}^{n \times n}$, $\mathcal{K}_{i,j} = K(X_i, X_j)$ so the training phase just operates on $\mathcal{K}$ to return an implicit representation of the separating hyperplane in the form of coefficients $\{\alpha_i\}_{i=1}^{n}$ and a threshold $\alpha_0$ used as follows:

> A future test point $x \in \mathbb{R}^D$, is deemed anomalous if it maps as $\phi(x)$ to *the wrong side* of the hyperplane, that is, if $f(x) \doteq \sum_{i=1}^{n} \alpha_i K(X_i, x) < \alpha_0$.

In other words, as in Euclidean spaces, $f(x)$ can be viewed as the projection of $\phi(x)$ onto a vector normal to the separating hyperplane, and the $\alpha_i$'s are coefficients determining this vector.

**Detection Time and Space.** It should be clear by now that computational complexity is determined by the number $\tilde{n} \leq n$ of nonzero $\alpha_i$'s. The corresponding data points $X_i$'s are called the *support vectors* and have to be kept in memory to estimate $f(x)$. Thus the OCSVM detector takes space $\tilde{n} \cdot (D+1)$, while the computation time for $f$ is $\Omega(\tilde{n} \times D)$. Unfortunately, it is often the case that $\tilde{n} = n$ or is of the same order, while the larger $n$, the more accurate the detector is.

### 3.2 Nyström and KJL Sketching

A main approach adopted recently to speedup training time, e.g., in the context of SVMs, is to reduce operations on the gram matrix $\mathcal{K} \in \mathbb{R}^{n \times n}$ by approximating it with a rank $d \ll n$ matrix $\mathcal{K}' \in \mathbb{R}^{n \times n}$ that might induce faster operations, while preserving much of the geometry induced by the kernel $K$ on the implicit mapping $\{\phi(X_i)\}_{i=1}^{n} \in \mathcal{H}$. These come in different forms under the name of Nyström and Sketching. In particular, in some implementations, we can view $\mathcal{K}'$ as inducing a new mapping $x \mapsto \phi'(x)$ for $\phi'(x) \in \mathbb{R}^d$, i.e., a low-dimensional mapping that preserves some geometry in $\mathcal{H}$.

Critically, as explained in the introduction, such $\phi'$ often no longer allows for linear separability from $0$ – i.e., using just one class in the training data – as in the case of the original OCSVM map $\phi$, since the remapped data $\{\phi'(X_i)\}_{i=1}^{d}$ no longer lies on the surface of a sphere (see Figure 2). However, cluster structures uncovered by the original $\phi$ are preserved since $\phi'$ preserves interpoint distances (see e.g., Calandriello & Rosasco (2018); Kpotufe & Sriperumbudur (2020)), which we build on in Section 4 below.

**The Embedding $\phi'$.** Crucially, in order to leverage cluster structures towards efficient outlier detection, we make the embedding $\phi'$ explicit – as opposed to operating on $\mathcal{K}'$ – and work directly in $\mathbb{R}^d$. This is based on recent reinterpretations of forms of Nyström and Sketching as low-dimensional projections (Yang et al., 2012; Kpotufe & Sriperumbudur, 2020). In both cases, let $S_m$ denote a random subsample of size $m \ll n$ of the training data $S_n \doteq \{X_i\}_{i=1}^{n}$ (w.l.o.g., we can let $S_m \doteq \{X_i\}_{i=1}^{m}$). Furthermore, for any subset of indices $I, J \subset \{1, \ldots, n\}$, let $\mathcal{K}_{I,J}$ denote the submatrix of $\mathcal{K}$ corresponding to rows in $I$, and columns in $J$. Then, for $I = \{1:m\}$ and $J = \{1:n\}$, we will consider the submatrices, $\mathcal{K}_{I,I} \in \mathbb{R}^{m \times m}$ – i.e., the gram matrix on $S_m$, and $\mathcal{K}_{I,J} \in \mathbb{R}^{m \times n}$, the gram submatrix of inner-products between $S_m$ and $S_n$.

- *Nyström.* Let $K_{I,I}^{-1}$ denote a rank $d$ pseudo-inverse of $K_{I,I}$; then setting $\mathcal{K}' = K_{I,J}^{\top} \cdot K_{I,I}^{-1} \cdot K_{I,J}$, the problem is to come up with $\phi' \in \mathbb{R}^d$ such that $\langle \phi'(X_i), \phi'(X_j) \rangle$ is exactly $\mathcal{K}_{i,j}'$. Recalling a bit of linear algebra, we can see that a suitable $\phi'$ can be defined as follows (Yang et al., 2012). Let $\Lambda \in \mathbb{R}^{d \times d}$ denote the diagonal matrix containing the top $d$ eigenvalues $\lambda_1, \ldots, \lambda_d$ of $\mathcal{K}_{I,I}$, and $V = [v_1, \ldots, v_d] \in \mathbb{R}^{m \times d}$ contains the corresponding (column) eigenvectors $v_i$'s. Now, for any $x \in \mathbb{R}^D$, let $K(x)$ denote $[K(x, X_1), \ldots, K(x, X_m)]^{\top}$, we then have

$$\phi'(x) \doteq P \cdot K(x), \text{ where we let } P \doteq \Lambda^{-1/2} \cdot V^{\top}. \tag{1}$$

We can verify that setting $K_{I,I}^{-1} = V \cdot \Lambda^{-1} \cdot V^{\top}$, indeed recovers $\mathcal{K}'$ as defined above.

- *KJL Sketching.* In general, Sketching consists of multiplying a gram matrix $\mathcal{K}$ (or $K_{I,I}$) by a matrix $Z$ with random entries. It was recently shown (Kpotufe & Sriperumbudur, 2020) that when $Z$ has i.i.d. $\mathcal{N}(0,1)$ Gaussian entries, sketching can be understood as a random projection operation in $\mathcal{H}$, leading to the following mapping $\phi' \in \mathbb{R}^D$. For any $x \in \mathbb{R}^D$, let $K(x)$ again denote the vector $[K(x, X_1), \ldots, K(x, X_m)]^\top$, and let $Z \in \mathbb{R}^{d \times m}$ with random $\mathcal{N}(0,1)$ entries. We then have:

$$\phi'(x) \doteq P \cdot K(x) \text{ where we let } P \doteq Z \cdot K_{I,I}. \tag{2}$$

**Embedding Time and Space.** Notice that in both cases of Nyström and KJL, we only have to retain $P \in \mathbb{R}^{d \times m}$ at testing time, along with the $m$ data points in $S_m$. Namely, the model $\phi'$ requires space complexity exactly $m \cdot (d + D)$. Similarly, the time complexity for evaluating $\phi'(x)$ is of order $m \cdot (d + D)$, so it does not depend on $n$.

As it turns out $m, d$ can be kept considerably smaller than $n$, while achieving the benefits of both methods. This is illustrated in Figure 3, on simulated data of size $n = 10000$, with two classes that are not easily clustered in $\mathbb{R}^D$, but which are clusterable not only in $\mathcal{H}$, but also after Nyström of KJL. In that simulation, we used $d = 2$ and $m = 100$. Similar small values are used for our experiments on real-world IoT data (see the experimental setup in Section 5.1).

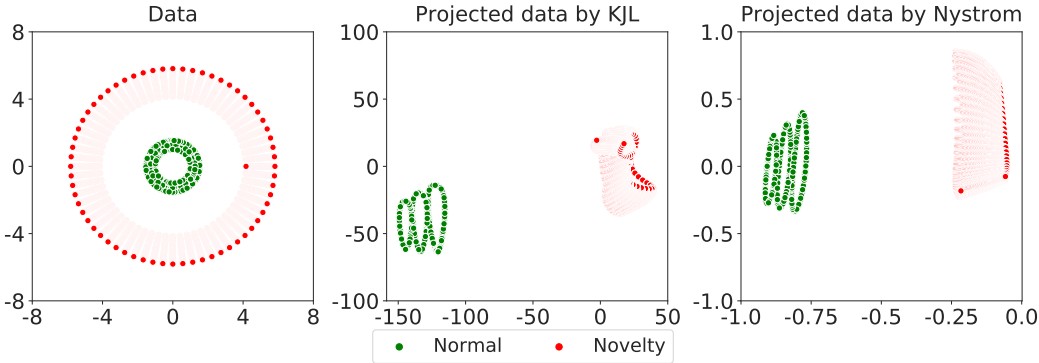

**Figure 3:** *Clusters after mapping $\phi'$: the simulation data* Cluster in Cluster *has 5000 points, shown before and after KJL/Nyström mapping. The KJL/Nyström mapping $\phi'$, shown on the right, retains the clusters uncovered by the initial kernel mapping $\phi$.*

# 4 Efficient Detection Procedures

Once the data is mapped to $\mathbb{R}^d$ as $\{\phi'(X_i)\}_{i=1}^n$ through Nyström or KJL, our next step is to learn an efficient model of the normal class embedded in $\mathbb{R}^d$. Recall that cluster structures are preserved, but not necessarily linear separability from 0 (see e.g., simulation of Figure 3 where the normal class is not necessarily linearly separable from the origin $0 \in \mathbb{R}^2$).

Looking somewhat ahead, this intuition is validated with the results of Figure 4 where we compare fitting a linear separator after KJL projection (denoted OC-KJL-SVM) to our proposed method (OC-KJL) soon to be described. The detection performance metric is the AUC, which is consistently higher for OC-KJL across datasets. A natural idea therefore is to flag future points as novelty if they fall far from clusters in $\{\phi'(X_i)\}_{i=1}^n$. This may be implemented in a number of ways via existing clustering procedures such as K-MEANS; however, it is then unclear how to efficiently and soundly evaluate what we might be *how far* a new point is from estimated clusters. We opt for a simple implementation consisting of fitting a *Gaussian*

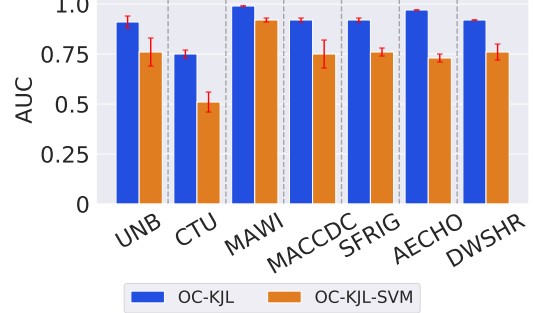

**Figure 4:** *OC-KJL (using a GMM) performs better than OC-KJL-SVM (using a linear separator).*

*Mixture Model* (GMM) to the normal remapped data $\{\phi'(X_i)\}_{i=1}^n$, with $k$ components that encode clusters; such a GMM is defined as a probability density of the form

$$f(z) = \sum_{l=1}^{k} \pi_l \cdot \mathcal{N}\left(z; \mu_l, \Sigma_l\right), \ \ for\ any\ z \in \mathbb{R}^d, \tag{3}$$

where $\mathcal{N}\left(z; \mu_l, \Sigma_l\right)$ denotes a Gaussian density with mean $\mu_l$ and covariance $\Sigma_l$ evaluated at $z$, and $\pi_l$'s denote the probability or mass of each cluster $l \in \{1, \ldots, k\}$ and sum up to 1. The GMM $f$ would have *modes* $\mu_l$, i.e., local maxima a.k.a. *high-density cores*, centered on clusters, as illustrated in Figure 2. An immediate advantage of employing GMM's as a clustering approach is that, once $f$ is learned, detection simply consists of flagging $x$ as a novelty if $f(\phi'(x))$ is smaller than a threshold $t$, i.e., if the remapped $\phi'(x)$ falls in a low-density region far from cluster means. In practice, such a threshold can be picked depending on the amount of tolerable false positive; for instance, to get at most 5% false positives, we set $t$ as the 95th quantile of $f$ values (in decreasing $f$ order) on the negative data, i.e., on the embedded normal data $\{\phi'(X_i)\}_{i=1}^n$. In our experiments below, we report the performance of detectors across all such thresholds choices, as captured by AUC (Sections 5.1 and 5.2).

**Meta Procedures.** The resulting OC-Nyström and OC-KJL approaches are summarized below. Given a Gaussian kernel $K$ with bandwidth $h$, embedding choices $m, d \ll$ *training size $n$*:

---

**Training:** Given normal data $\{X_i\}_{i=1}^n \in \mathbb{R}^D$ do:

- Embed $X_i$'s as $\phi'(X_i) \in \mathbb{R}^d$ via Nyström equation 1 or KJL equation 2;

- Parameter $k$ is passed in or is chosen via Quickshift++ (see paragraph below) on embedded data $\{\phi'(X_i)\}_{i=1}^n \in \mathbb{R}^d$;

- Estimate a GMM $f$ with $k$ components on $\{\phi'(X_i)\}_{i=1}^n$;

- **Return** GMM $f$ along with projection $\phi'$ (i.e., matrix $P$ and subsample $S_m$) $\quad\square$;

---

**Detection:** Given new $x \in \mathbb{R}^D$ and model $(\phi', f)$, do:

- Embed $x$ as $\phi'(x)$ into $\mathbb{R}^d$;

- Flag $x$ as novelty iff $f(\phi'(x)) \leq$ *threshold $t$* $\quad\square$

---

**Choice of Number of Components $k$.** As discussed earlier, we may automatically choose the number of components $k$ by first identifying the number of *high-density regions* in the mapped data $\{\phi'(X_i)\}$. This might be done in a number of ways, and we propose to use available density-mode estimators such as from the *Meanshift* family Comaniciu & Meer (1999); these are procedures that automatically identify the modes, i.e., local maxima, of the underlying data density, which in simple terms are just the regions of highest density in the data. In particular, in this work we employ a recent fast version of these mode estimators denoted *QuickShift++* Jiang et al. (2018), which automatically returns points in locally high-density regions of the data, *with no a priori knowledge of the number of such regions*, which we will identify with clusters. However, if labeled side data is available to cross-validate for the OCSVM or Nyström and KJL bandwidth parameter $h$, the same data can be used to choose $k$ (see Section 5.1).

**Detection Time and Space.** As described in Section 2.2., saving the model $\phi'$ takes space $m \cdot (d + D)$, while $f$ now takes additional space $k \cdot (d + d^2)$ for GMM parameters. As $m, d$ can be chosen small, detection time mostly depends on $k$; fortunately, as discussed in the introduction, $k$ can be chosen small (between 1 and 20 in our experiments) as clusters naturally correspond to the typically few modes of normal operation of IoT devices.

**OC-Nyström vs. OC-KJL.** As we will see in the results of Section 5.2, both procedures achieve our intended goal of efficiency while maintaining detection performance on par with that of OCSVM; while advantages vary across datasets, OC-Nyström tends to trade a bit of efficiency for better detection, as its embedding can require larger $k$ values.

# 5 Experimental Setup and Evaluation

## 5.1 Experimental Setup

**Data Sources.** We consider both publicly available traffic traces and traces collected on private consumer IoT devices. We aim to evaluate a representative set of devices, from multi-purpose devices, such as laptop PCs and Google Home, to less complex electronics and appliances with few modes of operations, such as smart cameras or smart fridges. Furthermore, we aim at a representative set of *novelties*, from benign novelties (new activity or a new device type) to novelties due to malicious activities (DDoS attack). Table 1 describes the datasets we used in the main paper, along with the associated types of novelty being detected. There are seven datasets in total, in which three of them are IoT datasets collected from three IoT devices deployed at the University of Chicago, and the remaining four are public datasets (i.e., CTU IoT, UNB IDS, MAWI, and MACCDC). Furthermore, the types of novelty vary from benign novelties (new activity or a new device type) to novelties due to malicious activities (DDoS attack).

**Table 1:** *Datasets' details.*

| Datasets | Description | Devices | Type of Novelty |
|---|---|---|---|
| Lab IoT SFRIG | Data traces are generated by a Samsung Fridge (SCam) with IP '192.168.202.43' in a private lab environment. It has two types of traffic traces labeled as *normal* when there is no human interaction and, *novel* when being operated by a human (such as opening the fridge). | One fridge | Novel activity |
| Lab IoT AECHO | Data traces are generated by an Amazon ECHO (AECHO) with IP '192.168.202.74' in a private lab environment. It has two types of traffic traces labeled as *normal* when there is no human interaction, and *novel* when being operated by a human (such as buying food by the ECHO). | One Amazon ECHO | Novel activity |
| Lab IoT DWSHR | Data traces are generated by a dishwasher (DWSHR) with IP '192.168.202.76' in a private lab environment. It has two types of traffic traces labeled as *normal* when there is no human interaction, and *novel* when being operated by a human (such as opening the dishwasher). We also add another novel traffic (such as buying food by an AECHO) collected from an Amazon ECHO (with IP '192.168.202.174') into *novel* to get a bigger testing set. | One dishwasher and one Amazon ECHO | Novel activity |
| CTU IoT (García, 2019) | Bitcoin-Mining and Botnet traffic traces generated by two Raspberries; we use Botnet traffic (with IP '192.168.1.196') as *normal* and Bitcoin-Mining traffic (with IP '192.168.1.195') as *novel*. | Two infected Raspberry Pis | Novel (infected) device |
| UNB IDS (Sharafaldin et al., 2018) | Normal traces are generated by one personal computer (PC) with IP address is '192.168.10.9'. Attack traces are generated by three PCs with IP addresses: '192.168.10.9', '192.168.10.14', and '192.168.10.15'. | Four PCs | DDoS attack |
| MAWI (Naga & Kaizaki, 2020) | Normal traffic are collected on July 01, 2020; we choose one kind of traffic generated by a PC with '203.78.7.165' as *normal*, and another kind of traffic generated by a PC with IP address '185.8.54.240' as *novel*. | Two PCs | Novel (normal) device |
| MACCDC (O'Brien et al., 2012) | Data traces are collected in 2012; we choose one kind of traffic generated by a PC with IP '192.168.202.79' as *normal* and one kind of traffic generated by a PC with IP '192.168.202.76' from another pcap as *novel*. | Two PCs | Novel (normal) device |

Although some of the devices that we test are multipurpose (as such might display more modalities than a special-purpose IoT device), including them allows us to test how well our approach scales. In particular, we will see that efficient detection is possible even in such cases, as even then $k \leq 20$ clusters suffice to maintain detection performance over these datasets, keeping $d = 5$ and $m = 100$.

**Representation: Flows.** Our unit of measurement consists of traffic flows, described below, i.e., as we aim to flag flows as normal or novel. We parse bidirectional flows from datasets in Table 1 using Scapy (Biondi, 2021) and extract interarrival times and packet sizes. Because certain devices can have arbitrarily long flows, we truncate each flow from a given dataset to have duration at most that of the 90th upper-percentile of flow durations in the dataset. Henceforth, a *flow* refers to these choices of flows involving truncation. We randomly split the obtained data into training, validation, and test sets of sizes detailed in Table B.1 in Appendix.

**Representation: Features.** Every flow is represented as a vector of the interarrival times between packets, i.e., in microseconds elapsed between consecutive packets, along with the size in bytes of each packet in the

flow (IAT+SIZE). We select these features as it results in competitive detection accuracy for OCSVM. This is illustrated, e.g., against 2 common alternative feature choices, namely, STATS and SAMP_SIZE, as shown in Table 2. A different choice, STATS+HEADER, corresponds to common statistics on flows, e.g., flow duration, mean, standard deviation and quantiles of packet sizes, in addition to packet header information (Yang et al., 2020) as explained in detail in Appendix E.2.

Results for the alternative features set STATS+HEADER are given in Appendix E.3, further demonstrating that our speedups generally hold over choices of data representation, as significant savings in time and space over baseline OCSVM remain. Such generality is expected because the main source of savings in both time and space stands, namely, the succinct finite-dimensional modeling of the infinite-dimensional representation inherent in OCSVM, made possible by the few modalities displayed by typical IoT devices.

**Implementation and Hyperparameters.** All detection procedures are implemented in Python, calling on the `scikit-learn` package for existing procedures such as OCSVM and GMM. While OCSVM training uses the standard `libsvm` package, we re-implemented its detection routines (as described in

**Table 2:** *Average AUCs for alternative features.*

| Dataset | MAWI | SFRIG | DWSHR |
|---|---|---|---|
| IAT+SIZE | $0.99 \pm 0.00$ | $0.93 \pm 0.00$ | $0.93 \pm 0.00$ |
| STATS | $1.00 \pm 0.00$ | $0.86 \pm 0.00$ | $0.65 \pm 0.00$ |
| SAMP_SIZE | $0.99 \pm 0.00$ | $0.93 \pm 0.00$ | $0.92 \pm 0.00$ |

Section 3.1) using `Numpy` to ensure fair, apples-to-apples execution time comparison with OC-Nyström and OC-KJL, which are implemented in `Numpy`, a Python library that calls on fast algebraic operations and parallel processing on multicore machines (Harris et al., 2020). The Nyström and KJL projections are implemented as described above; All the source codes can be seen at KJL

**Training Scenarios.** We consider two practical scenarios: one where some small amount of labeled *novelty* data is available to *validate* hyperparameter choice in a controlled lab environment, and one with no such labeled validation data, where we have to result in default choices of hyperparameters. Although each detection procedure may have many internal parameters, this distinction in scenarios only applies to two key choices of hyperparameters:

- *Kernel Bandwidth $h$.* For all methods, i.e., OCSVM, OC-Nyström, and OC-KJL, we use a Gaussian kernel of the form $K(x, x') \propto \exp(-\|x - x'\|^2/h^2)$, where the *bandwidth $h$* is to be picked as a quantile of $\binom{n}{2}$ distances between the $n$ training data points. In all our results, we consider 10 quantiles $[0.1, 0.2, \ldots, 0.9] \cup \{0.95\}$ of increasing interpoint distances.

- *Number of GMM components $k$.* As explained above, OC-Nyström and OC-KJL also require a choice of the number of GMM components to fit. We consider choices in the range $[1, 4, 6, 8, 10, 12, 14, 16, 18, 20]$. Thus the number of components or *clusters $k$* is capped at 20, as IoT devices are expected to display relatively few modes of operations, i.e., clusters of normal network activity.

  *For the results presented in the main text, the choice of $k$ is made by* QuickShift++*, and the resulting procedures are denoted* **OC-Nyström-QS** *and* **OC-KJL-QS**: these versions of our fast methods therefore only leave the choice of bandwidth $h$ and will be our main focus.

Next, we discuss how the above parameters are picked in each of the use cases or scenarios discussed above.

- **Minimal Tuning**. To simulate the first training scenario where some small amount of labeled novelty data is available, we subsample a small amount of the novelty data, which paired with equal amount of normal data is used to form a *validation set* to be used in hyperparameter choice. We then proceed to choose $h$ or $k$ (when Quickshift++ is not used) to minimize AUC over the validation data, so that these choices are independent of the random test set on which final results are reported.

- **No Tuning.** In this case, we choose the bandwidth $h$ by a common rule-of-thumb as the 0.3 quantile of increasing interpoint distances on the training data. The choice of the number of components $k$ is then always made by Quickshift++. We observed the same speedups under these settings, although these results are related to Appendix D for space.

**All Other Algorithmic Parameter Choices are Fixed.** We now describe all other choices inherent in our procedures, OC-Nyström and OC-KJL, and their variants OC-Nyström-QS and OC-KJL-QS.

- *Projection Parameters.* As discussed in Section 4, subsamples size $m$ and projection dimension $d$ are fixed to $m = 100$ and $d = 5$, choices which remarkably preserve detection performance across datasets and types of novelty, despite the considerable amount of *compression* they entail.

- *Quickshift++ Parameters.* We use the implementation of (Jiang et al., 2018; Jiang et al.), which requires internal parameters $\beta$ set to 0.9 (this performs density *smoothing*) and the number of neighbors set to $n^{2/3}$ (to build a *dense* neighborhood graph whose connectivity encodes high-density regions), two choices that work well across device datasets and types of novelty.

Here, due to variability in the data, Quickshift++ can often return too many *outlier* clusters (despite the conservative setting of its internal parameters). To remove those, we only retain *large* clusters, namely, the smallest number of clusters that account for at least 95% of the data if this number is less than 20; otherwise, we retain the 20 largest clusters discovered by Quickshift++.

*Gaussian Mixture Models Parameters.* We have the choice of using either *full* Gaussian covariances in fitting a GMM model to the projected data after KJL or Nyström or using only diagonal covariances for faster fitting – especially when operating in high dimensional settings – but at the usual cost of some loss in accuracy. Since GMMs are fit after projection to low dimension $d = 5$, it turns out that full Gaussian covariances are in fact efficient to fit in our case, so we only report results for full covariances.

When using Quickshift++, we initialize GMM with the clusters returned, i.e., local means and covariances of these clusters, and train till convergence.

**Computing Platforms.** We perform our experiments on two computing platforms: (1) a well-provisioned server, for the use case where all training and detection might occur offline; and (2) resource-constrained devices, specifically a Raspberry Pi and an NVidia Jetson Nano, corresponding to the use case where detection is to be real-time, local to the IoT device. Table A.1 in Appendix provides details.

## 5.2 Evaluation Metrics

**Detection Performance.** In novelty detection, there is a well-known tension between *false detection a.k.a. false positive rates* (FDR, i.e., the proportion of normal data wrongly flagged as novel) – and *true detection a.k.a. true positive rates* (TDR, i.e., the percentage of abnormal data rightly flagged as novel). Such tradeoffs are well captured by a Receiver Operating Characteristic (ROC) curve, which plots the detection rate TDR against the false alarm rate FDR as the detection threshold $t$ is varied from small to large; thus, the area under the ROC curve (**Area Under the Curve** (AUC)) when it is large, i.e., close to 1, indicates that good tradeoffs are achieved by the given detection approach. In contrast, AUC below 0.5 signals poor tradeoffs. AUC is therefore commonly adopted as a sensible measure of detection performance, as it captures tradeoffs under the complete range of detection choices.

In practice, a single threshold is chosen, driven by application-specific constraints, as one might prefer high TDR over low FDR, or vice versa (think of an infected medical device, e.g., a pacemaker, where high TDR would be preferred, vs. an infected smart home appliance, e.g., a toaster, where low FDR might be preferred). Large AUC, thus, indicates that the detector allows for good choices in any of these situations. For our proposed fast detectors, we will be interested in the fraction of AUC retained over OCSVM, i.e., the AUC of our detector divided by that of OCSVM.

**Training and Detection (or Testing) Time.** We will measure time as the *wall-clock time* taken by any of the methods for training (not-including data preprocessing into feature vectors, but inclusive of all actual training, i.e., modeling fitting) and *testing*, i.e., actual detection computations, on given machine environments (see Section 5.1 below) after a model is obtained. We report the *speedup*, the ratio of wall-clock time for OCSVM over that of our detector, separately for training and testing.

**Detection (or Testing) Space.** We report the space taken by the model returned by the detection procedure in kiloBytes. Namely, we report the minimal amount of information on the learned model to be saved towards future detection. That is, (1) support vectors and coefficients for OCSVM, and (2) projection parameters and GMM components for OC-Nyström and OC-KJL (with or without Quickshift++), all as described in Section 3.1 and 4. While memory usage depends on the programming language, under Python 3.7.3, memory usage is machine-independent, as Python enables porting across 64 or 32-bit machine architectures via its *pickling* process. All of our models are first trained on a 64-bit machine (Section 5.1).

**Averaging and Data Splitting.** To reduce uncertainty in reported results, we introduce repetitions in various stages of our experiments and report averages and standard deviations on performance metrics. For each dataset, first all flows (normal and abnormal) are preprocessed into the IAT+SIZE features, which creates at set of *normal* and novel data, from which we draw random subsamples. Experiments on each dataset follow the steps outlined below.

(i) Draw a subsample of size 600 to 2500 from normal data and a subsample of size 600 to 2500 from novelty data to form a test dataset of size 1200 to 5000. Exact sizes are given in Appendix Table B.1.

(ii) Repeat 5 times for accurate AUC:
- Draw a subsample of size $n = 10K$ from normal data to form the training data, except for MAWI ($n = 5.7K$).
- *If tuning:* draw a validation sample (1/4 test set size).
- Choose parameters $h, k$ as described in Section 5.1.
- Train with the choice of $h, k$ and save model on disk.
- Load and test model on Test data: repeat 100 times for accurate timing on machine (retain aggregate time).

For the baseline OCSVM, we report the average and standard deviation of performance metrics over the 5 repetitions. When reporting *speedups* for OC-Nyström and OC-KJL over OCSVM, we use the corresponding average performance of OCSVM, say $\mu$. In other words, if we observe AUCs $a_1, \ldots, a_5$ for OC-KJL, we report the mean of $a_1/\mu, \ldots a_5/\mu \pm$ the std of these ratios. We proceed similarly for time ratios.

## 6 Results

We focus in the main body on the first situation where some validation is available to tune model hyperparameters as described in Section 5.1. Results for the case of no tuning (as described in Section 5.1) reveal similar speedups in time and space as shown here and are given in detail in Appendix D. We further focus here on the OC-Nyström-QS and OC-KJL-QS—for fair comparison as OCSVM comes with a single parameter $h$—while results for OC-Nyström and OC-KJL (which require $h, k$ to be tuned) are given in Appendix C. As previously discussed in Section 5.1, all testing procedures are implemented in `Numpy` with parallelism turned on to take advantage of multicores.

**OCSVM Baseline Performance.** Table 3 shows OCSVM baseline performance. All training is performed on the server; testing is performed on all platforms. The table reports (1) AUC, same for all machines, since the same models and test data are used; (2) training time on the server; (3) test time for all three machines; and (4) test space, which is the same across all machines.

**Table 3:** *OCSVM baseline performance. Time is in milliseconds per 100 data points and space is in kiloBytes.*

| Dataset | | UNB | CTU | MAWI | MACCDC | SFRIG | AECHO | DWSHR |
|---|---|---|---|---|---|---|---|---|
| AUC | | $0.67 \pm 0.01$ | $0.66 \pm 0.02$ | $0.99 \pm 0.00$ | $0.85 \pm 0.00$ | $0.93 \pm 0.00$ | $0.90 \pm 0.00$ | $0.93 \pm 0.00$ |
| Server Train Time (ms) | | $93.93 \pm 4.75$ | $77.87 \pm 1.72$ | $82.59 \pm 1.53$ | $84.19 \pm 1.93$ | $80.83 \pm 7.72$ | $111.26 \pm 6.59$ | $110.53 \pm 5.77$ |
| Test Time (ms) | RSPI | $125.86 \pm 0.08$ | $118.19 \pm 4.54$ | $74.53 \pm 0.05$ | $124.58 \pm 0.22$ | $120.12 \pm 0.22$ | $125.43 \pm 0.22$ | $124.54 \pm 0.28$ |
| | NANO | $83.87 \pm 0.18$ | $86.71 \pm 0.82$ | $58.57 \pm 0.05$ | $86.89 \pm 0.80$ | $69.62 \pm 0.09$ | $88.80 \pm 0.08$ | $84.07 \pm 0.04$ |
| | Server | $19.84 \pm 0.16$ | $19.97 \pm 1.36$ | $11.22 \pm 0.14$ | $19.99 \pm 0.23$ | $16.15 \pm 0.20$ | $19.98 \pm 0.03$ | $19.81 \pm 0.01$ |
| Space (kB) | | $1763.53 \pm 0.45$ | $961.52 \pm 0.75$ | $2792.46 \pm 0.48$ | $1042.92 \pm 0.45$ | $641.58 \pm 0.10$ | $1441.75 \pm 0.14$ | $1202.39 \pm 0.30$ |

**AUC Retained.** As stated earlier, we now verify that our proposed methods manage to retain the accuracy of the baseline OCSVM and also do not sacrifice training efficiency. These are the results of Table 4. We

**Table 4:** *Retained AUC (method over OCSVM) and server train time speedup (OCSVM over method).*

| Dataset / Method | UNB | CTU | MAWI | MACCDC | SFRIG | AECHO | DWSHR |
|---|---|---|---|---|---|---|---|
| OC-KJL-QS: AUC Retained | $1.33 \pm 0.02$ | $1.05 \pm 0.03$ | $0.95 \pm 0.03$ | $1.03 \pm 0.01$ | $0.97 \pm 0.02$ | $1.07 \pm 0.00$ | $0.99 \pm 0.00$ |
| Train Speedup | $1.36 \pm 0.07$ | $1.10 \pm 0.02$ | $1.65 \pm 0.03$ | $1.14 \pm 0.03$ | $0.97 \pm 0.09$ | $1.18 \pm 0.07$ | $1.19 \pm 0.06$ |
| OC-Nyström-QS: AUC Retained | $1.37 \pm 0.02$ | $1.13 \pm 0.04$ | $0.90 \pm 0.03$ | $1.06 \pm 0.02$ | $0.96 \pm 0.01$ | $1.09 \pm 0.01$ | $0.98 \pm 0.01$ |
| Train Speedup | $1.32 \pm 0.07$ | $1.12 \pm 0.02$ | $1.74 \pm 0.03$ | $1.16 \pm 0.03$ | $0.93 \pm 0.09$ | $1.26 \pm 0.07$ | $1.17 \pm 0.06$ |

see that our detection methods largely retain the detection performance of OCSVM, often within a ratio of 1 or more, except in the case of MAWI, SFRFIG, and DWSHR – which are still high AUCs considering OCSVM's very good performance on these datasets. Moreover, for some datasets, such as UNB and CTU, all procedures manage to actually outperform OCSVM. It is likely that such higher performance is due to the additional regularization inherent in the dimension reduction performed by our methods. In particular, it is well known that dimension reduction approaches such as PCA have the added benefit of making clusters more salient by keeping cluster centers apart while reducing cluster diameters (see e.g., Sanjeev & Kannan (2001); Vempala & Wang (2004)). Here, we also re-emphasize that all methods presented are tuned fairly against the baseline OCSVM over the same bandwidth parameter $h$ as detailed in Section 5.1. Versions with Quickshift++, namely, OC-Nyström-QS and OC-KJL-QS, tend to achieve slightly smaller AUC compared to the non-Quickshift++ counterparts where the number of components $k$ is tuned by validation (see Table C.1), yet they also manage to maintain or sometimes outperform the baseline AUC of OCSVM.

**Training Time.** Our main goal is to maintain the training efficiency of OCSVM, and we see in Table 4 that our methods using Quickshift++ achieve this goal and even manage some minor speedup over OCSVM. We will see in Appendix C.1 that our methods without Quikshift++ achieve significantly faster training time, although with the added complexity of an additional tuning parameter $k$. OC-Nyström-QS's performance is similar to its non Quickshift++ counterpart, while at the same time it is more applicable in all scenarios, including when no validation data is available for tuning (results of Section D).

**Significant Savings in Detection Time and Space.** We report significant savings on detection time and space for all proposed variants of our approaches, which is the main motivation of this work. Table 5 and Table 6 present results for OC-KJL-QS and OC-Nyström-QS, while similar time and space savings under OC-KJL and OC-Nyström are presented in Appendix Tables C.2 and C.3.

**Table 5:** *OC-KJL-QS: Test time speedup (OCSVM over method) and space reduction (OCSVM over method).*

| Dataset | | UNB | CTU | MAWI | MACCDC | SFRIG | AECHO | DWSHR |
|---|---|---|---|---|---|---|---|---|
| Test Time Speedup | RSPI | $24.68 \pm 0.02$ | $24.22 \pm 0.93$ | $15.61 \pm 0.01$ | $27.56 \pm 0.05$ | $26.77 \pm 0.05$ | $26.38 \pm 0.05$ | $25.19 \pm 0.06$ |
| | NANO | $27.18 \pm 0.06$ | $31.67 \pm 0.30$ | $21.29 \pm 0.02$ | $33.95 \pm 0.31$ | $27.20 \pm 0.04$ | $30.44 \pm 0.03$ | $28.78 \pm 0.01$ |
| | Server | $26.33 \pm 0.21$ | $28.10 \pm 1.92$ | $17.17 \pm 0.21$ | $29.72 \pm 0.34$ | $24.15 \pm 0.30$ | $32.32 \pm 0.05$ | $30.94 \pm 0.01$ |
| Space Reduction | | $41.36 \pm 0.01$ | $36.30 \pm 0.03$ | $27.18 \pm 0.00$ | $38.21 \pm 0.02$ | $32.83 \pm 0.00$ | $40.73 \pm 0.00$ | $37.42 \pm 0.01$ |

**Table 6:** *OC-Nyström-QS: Test time speedup (OCSVM over method) and space reduction (OCSVM over method).*

| Dataset | | UNB | CTU | MAWI | MACCDC | SFRIG | AECHO | DWSHR |
|---|---|---|---|---|---|---|---|---|
| Test Time Speedup | RSPI | $22.86 \pm 0.02$ | $22.63 \pm 0.87$ | $15.54 \pm 0.01$ | $27.08 \pm 0.05$ | $26.48 \pm 0.05$ | $26.24 \pm 0.05$ | $25.56 \pm 0.06$ |
| | NANO | $24.73 \pm 0.05$ | $29.63 \pm 0.28$ | $21.53 \pm 0.02$ | $33.56 \pm 0.31$ | $27.39 \pm 0.04$ | $30.07 \pm 0.03$ | $29.14 \pm 0.01$ |
| | Server | $24.16 \pm 0.20$ | $25.90 \pm 1.77$ | $17.31 \pm 0.21$ | $29.44 \pm 0.33$ | $24.36 \pm 0.30$ | $31.73 \pm 0.05$ | $31.23 \pm 0.01$ |
| Space Reduction | | $39.99 \pm 0.01$ | $35.09 \pm 0.03$ | $27.21 \pm 0.00$ | $37.98 \pm 0.02$ | $32.97 \pm 0.00$ | $40.66 \pm 0.00$ | $37.63 \pm 0.01$ |

Our approaches are at least 15 times faster than OCSVM on every machine we considered: Nvidia Nano, Raspberry Pi, and the server. Speedups on Raspberry Pi and the server are most considerable, up to 20+ times faster than OCSVM on many datasets. The smaller amount of speedup that we observe on the Nano can be attributed to the relatively smaller amount of memory that this device has compared to the Raspberry Pi, which likely forces more memory swap operations as all test data is loaded at once. We also note that

unlike the Nano and Raspberry Pi, the server may have had more competing processes, yet even on the server, the trend of large speedups is observed across datasets. We see a small distinction between OC-KJL-QS and OC-Nyström-QS, whereby the former tends to achieve higher speedups on all machines for most datasets. As such both approaches seem to offer a tradeoff where, as per Table 4, the Nyström based approaches tend to achieve slightly higher AUC on most datasets.

In all cases, we observe significant space reductions, as our models can be stored upwards of 27 times less space and up to 41 times less than the baseline OCSVM model. This smaller memory footprint implies the possibility for a wider deployment than a conventional OCSVM, especially on memory-restricted devices such as on the embedded devices we evaluated. Although we focused much of our evaluation on memory-constrained devices, which is a common deployment scenario for IoT, the space efficiency of these models is important even in server settings, where a server might host large numbers of detection tools each dedicated to monitoring a given machine on client networks.

## 7 Conclusion

Because many IoT devices have a few well-defined operating regimes, it is possible to *model* their network flows in terms of relatively few *clusters* of activity under appropriate representations of the data. In this paper, we have extended the OCSVM approach, which has received much attention in IoT, to more efficient representations using projection and clustering. We have demonstrated that these procedures result in 15-40x improvements in both time and space without sacrificing detection accuracy across a wide range of novelty detection problems in IoT. These approaches, OC-Nyström-QS and OC-KJL-QS, are more widely applicable under practical use cases of novelty detection in IoT and in particular deployable not only on powerful servers but also on single-board computing devices with more limited memory and computing resources. Our evaluation of these techniques have also exposed some clear tradeoffs: when minimally tuned with a few labeled data, OC-Nyström-QS tends to achieve higher detection performance than OC-KJL-QS at the cost of some increase in computation time. As explained earlier in Section 6, additional results for other variants and use cases with no tuning shown in Appendix show similar significant speedups.

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

## A    Computing Platforms

Table. A.1 shows all three machines' information (i.e., Large server, Raspberry Pi, and Nvidia Nano), which includes operating system, CPU, memory, storage, programming language, and a scientific computing package (Numpy). Note that 'lscpu' command is used to get CPU and CPU cache information. In addition, for Raspberry Pi, we get the cache information from the ARM official document of the Cortex-72 processor (ARM, 2016).

**Table A.1:** *We train on server and test on all 3 machines.*

| Machine | Description |
|---|---|
| Large Server | 64-bit, running Debian GNU/Linux 9 (stretch) with Intel(R) Xeon(R) processor (32 CPU Cores, 1200-3400 MHz each), 100GB memory, and 2TB disk. |
| | Caches: 3 layers of CPU caches (i.e., 32KiB L1d and L1i, 256KiB L2, and 20MB L3). |
| | Programming language: Python 3.7.9. |
| | Numpy 1.19.2 (install from the numpy wheel that includes an OpenBLAS implementation of the BLAS and LAPACK linear algebra APIs (Oliphant, 2005)). |
| Raspberry Pi | 32-bit, running Raspbian GNU/Linux 10 (buster) with Cortex-A72 processor (4 CPU cores, 600-1500 MHz each), 8GB memory, and 27GB disk. |
| | Caches: 2 layers of CPU caches (i.e., 32KiB L1d, 48KiB L1i, and 1MiB L2 (ARM, 2016)). |
| | Programming language: Python 3.7.3 with '–enable-optimizations' option. |
| | Numpy 1.18.2 (install from the numpy source codes that searches for BLAS and LAPACK dynamic link libraries at build time as influenced by the system environment variables (Oliphant, 2005)). |
| Nvidia Nano | 64-bit, running Ubuntu 18.04.5 LTS (Bionic Beaver) with Cortex-A57 processor (4 CPU cores, 102-1479 MHz each), 4GB memory, and 30GB disk. |
| | Caches: 2 layers of CPU caches (i.e., 32KiB L1d, 48KiB L1i, and 2MiB L2) |
| | Programming language: Python 3.7.3 with '–enable-optimizations' option. |
| | Numpy 1.18.2 (install from the source codes (Oliphant, 2005)). |

## B    Dataset Sizes and Dimensions

Table B.1 shows Train set, Validation (Val.) set, Test set, and feature dimensions. The size of validation sets is always 1/4 of the corresponding test set sizes. In Validation set and Test set, the number of normal and abnormal data is equal. Moreover, the IAT+SIZE dimension of each dataset is less than 50, except for MAWI (its dimension is 121).

**Table B.1:** *Dataset sizes (# of data points) and dimensions.*

| Dataset | UNB | CTU | MAWI | MACCDC | SFRIG | AECHO | DWSHR |
|---|---|---|---|---|---|---|---|
| Train Set | 10,000 | 10,000 | 5,720 | 10,000 | 10,000 | 10,000 | 10,000 |
| Val. Set | 462 | 1,250 | 1,040 | 1,250 | 1,250 | 280 | 508 |
| Test Set | 1,854 | 5,000 | 4,160 | 5,000 | 5,000 | 1,120 | 2,032 |
| Dimensions | 43 | 23 | 121 | 25 | 15 | 35 | 29 |

## C    OC-Nyström and OC-KJL Results (no Quickshift++)

In the main paper body, we left out some of the detection time and space savings results for the OC-KJL and OC-Nyström variants (which do not use Quickshift++ for automatic cluster-number identification). Here we consider the effect of additionally tuning the number of GMM components rather than selecting them automatically via Quickshift++. The main message here is that not much is lost in AUC by automatically choosing $k$ via Quickshift++.

### C.1    Retained AUC and Training Efficiency

Table C.1 compares AUC and training times of OC-KJL and OC-Nyström to that of the baseline OCSVM.

**Table C.1:** *Retained AUC (method over OCSVM) and server train time speedup (OCSVM over method).*

| Dataset \ Method | UNB | CTU | MAWI | MACCDC | SFRIG | AECHO | DWSHR |
|---|---|---|---|---|---|---|---|
| OC-KJL: AUC Retained | 1.36 ± 0.04 | 1.13 ± 0.04 | 0.99 ± 0.01 | 1.08 ± 0.01 | 0.99 ± 0.01 | 1.08 ± 0.00 | 0.99 ± 0.00 |
| Train Speedup | 2.75 ± 0.14 | 2.44 ± 0.05 | 4.13 ± 0.08 | 2.47 ± 0.06 | 2.22 ± 0.21 | 2.84 ± 0.17 | 2.75 ± 0.14 |
| OC-Nyström: AUC Retained | 1.41 ± 0.02 | 1.19 ± 0.03 | 0.99 ± 0.01 | 1.07 ± 0.03 | 0.97 ± 0.01 | 1.09 ± 0.00 | 0.99 ± 0.01 |
| Train Speedup | 3.14 ± 0.16 | 2.59 ± 0.06 | 3.68 ± 0.07 | 2.60 ± 0.06 | 2.17 ± 0.21 | 2.94 ± 0.17 | 2.72 ± 0.14 |

**AUC Retained.** We see that our detection methods, OC-Nyström and OC-KJL, retain the detection performance of OCSVM, all within a ratio of 1 or more, except in the case of MAWI, SFRFIG, and DWSHR – which are still high AUCs considering OCSVM's high accuracy on MAWI, SFRIG, and DWSHR (Table 3). Moreover, for some datasets, such as UNB and CTU, all procedures manage to actually outperform OCSVM. It is likely that such higher performance is due to the additional regularization inherent in the dimension reduction performed by our methods.

**Training Time.** Although our original goal was just to maintain the training efficiency of OCSVM, especially considering the various additional steps inherent in our methods, our methods without Quickshift++ in fact achieve speedup – factors of 2-4 in some cases – over OCSVM training time which involves more expensive model fitting steps.

## C.2 Significant Savings in Detection Time and Space

Tables C.2 and C.3 presents results on detection time and space savings for both OC-KJL and OC-Nyström.

**Table C.2:** *OC-KJL: Test time speedup (OCSVM over method) and space reduction (OCSVM over method).*

| Dataset | | UNB | CTU | MAWI | MACCDC | SFRIG | AECHO | DWSHR |
|---|---|---|---|---|---|---|---|---|
| Test Time Speedup | RSPI | 26.66 ± 0.02 | 22.81 ± 0.88 | 13.25 ± 0.01 | 25.91 ± 0.05 | 26.20 ± 0.05 | 24.52 ± 0.04 | 25.07 ± 0.06 |
| | NANO | 29.66 ± 0.06 | 29.90 ± 0.28 | 18.24 ± 0.02 | 31.45 ± 0.29 | 27.13 ± 0.04 | 27.36 ± 0.02 | 28.38 ± 0.01 |
| | Server | 28.80 ± 0.23 | 26.41 ± 1.80 | 14.73 ± 0.18 | 28.19 ± 0.32 | 23.63 ± 0.29 | 29.11 ± 0.05 | 30.48 ± 0.01 |
| Space Reduction | | 42.55 ± 0.01 | 35.12 ± 0.03 | 26.52 ± 0.00 | 36.74 ± 0.02 | 32.58 ± 0.00 | 39.20 ± 0.00 | 37.13 ± 0.01 |

**Table C.3:** *OC-Nyström: Test time speedup (OCSVM over method) and space reduction (OCSVM over method).*

| Dataset | | UNB | CTU | MAWI | MACCDC | SFRIG | AECHO | DWSHR |
|---|---|---|---|---|---|---|---|---|
| Test Time Speedup | RSPI | 27.17 ± 0.02 | 24.30 ± 0.93 | 12.59 ± 0.01 | 27.02 ± 0.05 | 24.87 ± 0.05 | 25.33 ± 0.04 | 24.49 ± 0.06 |
| | NANO | 29.98 ± 0.07 | 31.92 ± 0.30 | 17.14 ± 0.02 | 33.62 ± 0.31 | 25.88 ± 0.03 | 28.24 ± 0.02 | 27.94 ± 0.01 |
| | Server | 28.31 ± 0.23 | 27.60 ± 1.88 | 13.95 ± 0.17 | 29.71 ± 0.34 | 23.22 ± 0.29 | 30.24 ± 0.05 | 29.91 ± 0.01 |
| Space Reduction | | 42.79 ± 0.01 | 36.55 ± 0.03 | 26.25 ± 0.00 | 37.98 ± 0.02 | 31.60 ± 0.00 | 39.82 ± 0.00 | 36.89 ± 0.01 |

**Testing Time Speedup.** We observe speedups of at least 13 times over the baseline OCSVM detection times across all machines and datasets.

**Space Reduction.** As before, space reductions are significant w.r.t. the baseline OCSVM from 26 to 42+ times less space than required by the baseline.

# D Results under No Tuning

We now consider the scenario where no validation data is available to tune any of the procedures, i.e., in choosing the bandwidth parameter $h$. While in general it is preferable to perform some minimal tuning before deployment, in practice, it may be difficult to obtain labeled data for the types of novel activities of interest that commonly arise in actual deployment environments.

In the practice of novelty detection with OCSVM, when no labeled data is available, various rule-of-thumbs are used, a popular one being to pick $h$ as a quantile of inter-point distances. For uniformity, as explained in Section 5.1, here we pick $h$ for all methods, as the 30th percentile of increasing inter-point distances in the training data.

Naturally, detection performance suffers w.r.t. that of a tuned procedure for any of the methods. Furthermore, since the choice of bandwidth affects the learned model, it is to be expected that time and space comparisons would also differ from that under minimal tuning as in the previous Section 6.

### D.1 Baseline OCSVM Performance

Table D.1 shows the performance of the baseline OCSVM. We observe a decrease in AUC for most datasets, most considerably for UNB and CTU, which already were hard datasets even under tuning (Table 3). Interestingly, MAWI, MACCDC, SFRIG, and DWSHR still admit high AUCs even without tuning, attesting to the general appeal of OCSVM as an adaptable and robust novelty detection approach. Although AECHO and MACCDC dropped in accuracy (from 0.85+), they still retain reasonable accuracies with AUC's at 0.78.

**Table D.1:** *OCSVM baseline performance,* **no tuning**. *Time is in ms per 100 datapoints and space is in kB.*

| Dataset | | UNB | CTU | MAWI | MACCDC | SFRIG | AECHO | DWSHR |
|---|---|---|---|---|---|---|---|---|
| AUC | | $0.60 \pm 0.01$ | $0.57 \pm 0.01$ | $0.98 \pm 0.00$ | $0.78 \pm 0.00$ | $0.85 \pm 0.00$ | $0.78 \pm 0.00$ | $0.87 \pm 0.01$ |
| Server Train Time (ms) | | $94.23 \pm 1.66$ | $81.47 \pm 2.39$ | $78.48 \pm 0.93$ | $84.72 \pm 1.01$ | $85.15 \pm 2.99$ | $117.65 \pm 6.46$ | $117.29 \pm 8.81$ |
| Test Time (ms) | RSPI | $124.54 \pm 0.12$ | $124.12 \pm 0.35$ | $74.48 \pm 0.64$ | $124.17 \pm 0.59$ | $123.70 \pm 0.36$ | $126.31 \pm 0.95$ | $125.46 \pm 0.72$ |
| | NANO | $89.30 \pm 0.11$ | $86.32 \pm 0.17$ | $50.71 \pm 0.05$ | $86.70 \pm 0.21$ | $83.50 \pm 0.17$ | $88.85 \pm 0.02$ | $88.00 \pm 0.09$ |
| | Server | $19.20 \pm 0.13$ | $19.42 \pm 0.22$ | $9.76 \pm 0.13$ | $19.10 \pm 0.27$ | $18.62 \pm 0.15$ | $19.95 \pm 0.02$ | $19.72 \pm 0.09$ |
| Space (kB) | | $1761.21 \pm 0.17$ | $961.40 \pm 0.23$ | $2798.32 \pm 0.78$ | $1041.34 \pm 0.13$ | $641.07 \pm 0.05$ | $1441.75 \pm 0.14$ | $1201.38 \pm 0.32$ |

### D.2 Retained AUC and Training Efficiency

Table D.2 compares AUC and training times of OC-KJL-QS and OC-Nyström-QS to that of the baseline OCSVM, using the exact same default choice of bandwidth $h$ as OCSVM.

**Table D.2: No tuning.** *Retained AUC (method over OCSVM) and train time speedup (OCSVM over method).*

| Dataset / Method | UNB | CTU | MAWI | MACCDC | SFRIG | AECHO | DWSHR |
|---|---|---|---|---|---|---|---|
| OC-KJL-QS: AUC Retained | $1.49 \pm 0.02$ | $1.20 \pm 0.05$ | $0.14 \pm 0.00$ | $0.90 \pm 0.14$ | $0.98 \pm 0.03$ | $1.20 \pm 0.03$ | $0.95 \pm 0.02$ |
| Train Speedup | $1.32 \pm 0.02$ | $1.14 \pm 0.03$ | $1.37 \pm 0.02$ | $1.16 \pm 0.01$ | $0.98 \pm 0.03$ | $1.29 \pm 0.07$ | $1.26 \pm 0.09$ |
| OC-Nyström-QS: AUC Retained | $1.54 \pm 0.03$ | $1.19 \pm 0.10$ | $0.17 \pm 0.00$ | $0.88 \pm 0.17$ | $0.77 \pm 0.11$ | $1.18 \pm 0.00$ | $0.90 \pm 0.05$ |
| Train Speedup | $1.32 \pm 0.02$ | $1.14 \pm 0.03$ | $1.47 \pm 0.02$ | $1.14 \pm 0.01$ | $0.97 \pm 0.03$ | $1.26 \pm 0.07$ | $1.22 \pm 0.09$ |

**AUC Retained.** OC-KJL-QS and OC-Nyström-QS manage to retain the AUC of OCSVM on most datasets. However, on MAWI, neither OC-Nyström-QS nor OC-KJL-QS does well, arriving at just a fraction of the baseline AUC. MACCDC and SFRIG appear to cause problems for OC-Nyström-QS under the default $h$ setting.

**Training Time.** As before, training time remains competitive with that of OCSVM with some significant reduction in time for instance in the case of UNB, AECHO, and DWSHR.

### D.3 Significant Savings in Detection Time and Space

Tables D.3 and D.4 present results on detection time and space savings for both OC-KJL-QS and OC-Nyström-QS, again with the same default choice of bandwidth $h$ as OCSVM. The trends on savings are similar, but in fact even better than those under minimal tuning of these 3 methods.

**Table D.3:** *OC-KJL-QS,* **no tuning**. *Test time speedup (OCSVM over method) and space reduction (OCSVM over method).*

| Dataset | | UNB | CTU | MAWI | MACCDC | SFRIG | AECHO | DWSHR |
|---|---|---|---|---|---|---|---|---|
| Test Time Speedup | RSPI | 24.30 ± 0.02 | 25.81 ± 0.07 | 12.39 ± 0.11 | 27.07 ± 0.13 | 27.37 ± 0.08 | 25.66 ± 0.19 | 25.40 ± 0.15 |
| | NANO | 28.94 ± 0.04 | 32.18 ± 0.07 | 15.99 ± 0.02 | 34.43 ± 0.08 | 34.21 ± 0.07 | 29.97 ± 0.01 | 30.88 ± 0.03 |
| | Server | 25.80 ± 0.18 | 28.28 ± 0.33 | 13.38 ± 0.18 | 28.85 ± 0.41 | 28.30 ± 0.23 | 32.34 ± 0.03 | 31.49 ± 0.15 |
| Space Reduction | | 41.30 ± 0.00 | 36.50 ± 0.01 | 26.28 ± 0.01 | 38.43 ± 0.00 | 33.57 ± 0.00 | 40.67 ± 0.00 | 37.80 ± 0.01 |

**Table D.4:** *OC-Nyström-QS,* **no tuning**. *Test time speedup (OCSVM over method) and space reduction (OCSVM over method).*

| Dataset | | UNB | CTU | MAWI | MACCDC | SFRIG | AECHO | DWSHR |
|---|---|---|---|---|---|---|---|---|
| Test Time Speedup | RSPI | 22.70 ± 0.02 | 24.50 ± 0.07 | 12.38 ± 0.11 | 26.31 ± 0.13 | 26.79 ± 0.08 | 23.75 ± 0.18 | 24.79 ± 0.14 |
| | NANO | 26.36 ± 0.03 | 30.28 ± 0.06 | 15.97 ± 0.02 | 33.22 ± 0.08 | 33.29 ± 0.07 | 27.37 ± 0.01 | 29.92 ± 0.03 |
| | Server | 23.06 ± 0.16 | 26.89 ± 0.31 | 13.38 ± 0.18 | 28.06 ± 0.40 | 28.37 ± 0.23 | 29.33 ± 0.03 | 30.60 ± 0.15 |
| Space Reduction | | 39.94 ± 0.00 | 35.34 ± 0.01 | 26.28 ± 0.01 | 37.78 ± 0.00 | 33.02 ± 0.00 | 39.28 ± 0.00 | 37.31 ± 0.01 |

**Testing Time Speedup.** We observe speedups of at least 12 times over the baseline OCSVM detection times across all machines and datasets.

Finally, we again observe the trend where OC-KJL-QS manages faster times than OC-Nyström-QS in most cases, especially on Raspberry Pi and the server.

**Space Reduction.** As before, space reductions are significant w.r.t. the baseline OCSVM from 26 to 41+ times less space than required by the baseline.

# E   Alternative Features

## E.1   SAMP-SIZE Features Description

SAMP-SIZE: a flow is partitioned into small time intervals of equal length, and the total packet size (i.e., byte count) in each interval is recorded; thus, a flow is represented as a time series of byte counts in small time intervals. Here, we obtain time intervals according to different quantiles (i.e., [0.1, 0.2, 0.3, 0.4, 0.5, 0.6, 0.7, 0.8, 0.9, 0.95]) of flow durations. To ensure that each sample has the same dimension $D$, we select $D$ for all flows as the 90th percentile of all *flow lengths* in the dataset (here, *flow length* stands for the number of packets a flow – not its duration).

Now for any given flow, if the number of fixed time intervals in the flow is less than $D$, we append 0's to arrive at a vector of dimension $D$. If instead the number of fixed time intervals is greater than $D$, we truncate the resulting vector representation down to dimension $D$.

## E.2   STATS+HEADER Features Description

STATS+HEADER: a set of statistical quantities compiled from a flow. In particular, we choose 10 of the most common such statistics in the literature (see e.g., Moore et al. (2013)), namely, flow duration, number of packets sent per second, number of bytes per second, and the following statistics on packet sizes (in bytes) in a flow: mean, standard deviation, the first to third quantiles, the minimum, and maximum. Also, We incorporate packet header information (i.e., Time to Live (TTL) and TCP flags (FIN, SYN, RST, PSH, ACK, URG, ECE, and CWR) into the STATS to form the STATS+HEADER feature.

**Table E.1:** *OCSVM performance with STATS+HEADER. Time is in ms per 100 datapoints and space is in KB.*

| Dataset | | UNB | CTU | MAWI | MACCDC | SFRIG | AECHO | DWSHR |
|---|---|---|---|---|---|---|---|---|
| AUC | | $0.60 \pm 0.01$ | $0.61 \pm 0.00$ | $1.00 \pm 0.00$ | $0.76 \pm 0.00$ | $0.86 \pm 0.00$ | $0.98 \pm 0.00$ | $0.65 \pm 0.00$ |
| Server Train Time (ms) | | $77.20 \pm 1.75$ | $94.54 \pm 2.73$ | $64.63 \pm 0.82$ | $92.23 \pm 2.25$ | $88.69 \pm 0.90$ | $100.58 \pm 2.78$ | $92.59 \pm 2.93$ |
| Test Time (ms) | RSPI | $123.90 \pm 0.39$ | $119.29 \pm 3.31$ | $72.39 \quad \pm 0.08$ | $114.20 \pm 0.33$ | $124.53 \pm 0.57$ | $123.42 \pm 0.34$ | $125.70 \pm 0.22$ |
| | NANO | $76.86 \pm 0.26$ | $87.13 \pm 0.31$ | $54.33 \pm 0.01$ | $87.22 \pm 0.06$ | $82.89 \pm 0.05$ | $78.33 \pm 0.04$ | $83.53 \pm 0.06$ |
| | Server | $17.60 \pm 0.38$ | $19.08 \pm 0.36$ | $11.17 \pm 0.16$ | $19.21 \pm 0.18$ | $19.09 \pm 0.23$ | $19.45 \pm 0.02$ | $19.83 \pm 0.02$ |
| Space (kB) | | $1655.41 \pm 2.72$ | $1240.96 \pm 0.16$ | $1831.75 \pm 0.00$ | $1280.82 \pm 0.13$ | $1084.04 \pm 0.22$ | $1482.07 \pm 0.30$ | $1363.32 \pm 0.28$ |

### E.3 Results under STATS+HEADER

#### E.3.1 Results Under Minimal Tuning

Table. E.1 shows the baseline results obtained by OCSVM under minimal turning. Accuracies of the baseline OCSVM are similar to those using the features of IAT+SIZE in the main paper and are reported for completion.

Similar to the case of IAT+SIZE features, both OC-KJL and OC-Nyström with or without Quickshift++ retain the AUC of the baseline OCSVM (on UNB and DWSHR, our procedures even get higher AUCs than OCSVM). Moreover, these procedures have 2-4 times train time speedup. More details are shown in Table E.2.

**Table E.2:** *Retained AUC (method over OCSVM) and server train time speedup (OCSVM over method) with STATS+HEADER.*

| Method \ Dataset | UNB | CTU | MAWI | MACCDC | SFRIG | AECHO | DWSHR |
|---|---|---|---|---|---|---|---|
| OC-KJL: AUC Retained | $1.33 \pm 0.07$ | $0.94 \pm 0.08$ | $1.00 \pm 0.00$ | $0.94 \pm 0.01$ | $1.01 \pm 0.01$ | $1.00 \pm 0.01$ | $1.18 \pm 0.03$ |
| Train Speedup | $2.63 \pm 0.06$ | $3.11 \pm 0.09$ | $3.02 \pm 0.04$ | $2.98 \pm 0.07$ | $2.63 \pm 0.03$ | $3.33 \pm 0.09$ | $2.52 \pm 0.08$ |
| OC-KJL-QS: AUC Retained | $1.13 \pm 0.05$ | $0.87 \pm 0.04$ | $0.98 \pm 0.01$ | $0.89 \pm 0.02$ | $0.98 \pm 0.01$ | $0.98 \pm 0.01$ | $1.05 \pm 0.08$ |
| Train Speedup | $1.02 \pm 0.02$ | $1.34 \pm 0.04$ | $1.40 \pm 0.02$ | $1.39 \pm 0.03$ | $1.31 \pm 0.01$ | $1.34 \pm 0.04$ | $1.18 \pm 0.04$ |
| OC-Nyström: AUC Retained | $1.45 \pm 0.02$ | $1.02 \pm 0.05$ | $0.99 \pm 0.00$ | $0.94 \pm 0.05$ | $0.96 \pm 0.02$ | $0.99 \pm 0.01$ | $1.18 \pm 0.04$ |
| Train Speedup | $2.45 \pm 0.06$ | $3.19 \pm 0.09$ | $3.24 \pm 0.04$ | $3.15 \pm 0.08$ | $3.15 \pm 0.03$ | $3.17 \pm 0.09$ | $2.60 \pm 0.08$ |
| OC-Nyström-QS: AUC Retained | $1.42 \pm 0.03$ | $0.94 \pm 0.04$ | $0.96 \pm 0.03$ | $0.84 \pm 0.04$ | $0.94 \pm 0.01$ | $1.00 \pm 0.01$ | $1.06 \pm 0.03$ |
| Train Speedup | $1.03 \pm 0.02$ | $1.29 \pm 0.04$ | $1.46 \pm 0.02$ | $1.48 \pm 0.04$ | $1.31 \pm 0.01$ | $1.39 \pm 0.04$ | $1.19 \pm 0.04$ |

We also see that these methods under the alternative features attain significant detection time speedups over OCSVSM, which are shown in Tables E.3 and E.4.

**Testing Time Speedup.** We observe speedups of at least 16 times over the baseline OCSVM detection times across all machines and datasets.

**Space Reduction.** As before, space reductions are significant w.r.t. the baseline OCSVM from 26 to 41+ times less space than required by the baseline.

#### E.3.2 Results Under No Tuning

OCSVM results under no tuning for STATS+HEADER features are presented in Table E.5. As with the case of our preferred features of IAT+SIZE, we observe a significant decrease in AUC w.r.t. the tuned OCSVM case.

We also get similar significant test time speedup and space reduction results for both methods as shown in Tables E.7 and E.8. This goes to show that the reductions inherent in our approach are likely not tied to feature representations of the networking data.

**Table E.3:** *OC-KJL-QS: Test time speedup (OCSVM over method) and space reduction (OCSVM over method) with STATS+HEADER.*

| Dataset | | UNB | CTU | MAWI | MACCDC | SFRIG | AECHO | DWSHR |
|---|---|---|---|---|---|---|---|---|
| Test Time Speedup | RSPI | 23.81 ± 0.07 | 28.17 ± 0.78 | 16.45 ± 0.02 | 27.00 ± 0.08 | 29.66 ± 0.14 | 24.92 ± 0.07 | 26.83 ± 0.05 |
| | NANO | 25.74 ± 0.09 | 36.96 ± 0.13 | 21.17 ± 0.01 | 35.58 ± 0.02 | 35.52 ± 0.02 | 25.93 ± 0.01 | 33.73 ± 0.02 |
| | Server | 27.23 ± 0.59 | 30.73 ± 0.58 | 17.37 ± 0.25 | 29.35 ± 0.27 | 30.53 ± 0.37 | 29.77 ± 0.03 | 36.73 ± 0.03 |
| Space Reduction | | 39.72 ± 0.07 | 40.88 ± 0.01 | 26.49 ± 0.00 | 40.72 ± 0.00 | 40.07 ± 0.01 | 40.39 ± 0.01 | 40.82 ± 0.01 |

**Table E.4:** *OC-Nyström-QS: Test time speedup (OCSVM over method) and space reduction (OCSVM over method) with STATS+HEADER.*

| Dataset | | UNB | CTU | MAWI | MACCDC | SFRIG | AECHO | DWSHR |
|---|---|---|---|---|---|---|---|---|
| Test Time Speedup | RSPI | 23.79 ± 0.07 | 27.04 ± 0.75 | 16.43 ± 0.02 | 27.22 ± 0.08 | 29.88 ± 0.14 | 25.11 ± 0.07 | 28.03 ± 0.05 |
| | NANO | 25.66 ± 0.09 | 35.34 ± 0.13 | 21.14 ± 0.01 | 35.17 ± 0.02 | 35.65 ± 0.02 | 25.24 ± 0.01 | 34.63 ± 0.02 |
| | Server | 27.08 ± 0.59 | 29.11 ± 0.55 | 17.34 ± 0.25 | 29.16 ± 0.27 | 30.67 ± 0.37 | 28.76 ± 0.03 | 37.39 ± 0.03 |
| Space Reduction | | 39.70 ± 0.07 | 40.33 ± 0.01 | 26.48 ± 0.00 | 40.64 ± 0.00 | 40.12 ± 0.01 | 39.94 ± 0.01 | 41.48 ± 0.01 |

**Table E.5:** *OCSVM performance with STATS+HEADER, **no tuning**. Time is in ms per 100 datapoints and space is in kB.*

| Dataset | | UNB | CTU | MAWI | MACCDC | SFRIG | AECHO | DWSHR |
|---|---|---|---|---|---|---|---|---|
| AUC | | 0.15 ± 0.00 | 0.41 ± 0.00 | 0.99 ± 0.00 | 0.50 ± 0.00 | 0.79 ± 0.00 | 0.96 ± 0.00 | 0.38 ± 0.01 |
| Server Train Time (ms) | | 92.28 ± 2.55 | 89.55 ± 2.28 | 62.00 ± 2.11 | 96.15 ± 1.96 | 89.70 ± 1.70 | 95.50 ± 2.22 | 97.06 ± 2.16 |
| Test Time (ms) | RSPI | 124.86 ± 0.96 | 123.78 ± 0.82 | 70.74 ± 0.36 | 124.65 ± 0.10 | 123.51 ± 0.23 | 127.09 ± 1.05 | 124.99 ± 0.49 |
| | NANO | 89.86 ± 0.90 | 87.65 ± 0.31 | 46.35 ± 0.04 | 86.49 ± 0.14 | 84.04 ± 0.19 | 88.14 ± 0.07 | 85.96 ± 0.08 |
| | Server | 19.91 ± 0.65 | 19.32 ± 0.36 | 10.00 ± 0.05 | 19.61 ± 0.20 | 18.91 ± 0.23 | 20.82 ± 0.01 | 19.91 ± 0.02 |
| Space (kB) | | 1641.24 ± 0.26 | 1241.11 ± 0.12 | 1832.14 ± 0.31 | 1282.76 ± 0.54 | 1081.45 ± 0.22 | 1481.54 ± 0.12 | 1361.80 ± 0.24 |

**Testing Time Speedup.** We observe speedups of at least 12 times over the baseline OCSVM detection times across all machines and datasets.

**Space Reduction.** As before, space reductions are significant w.r.t. the baseline OCSVM from 25 to 43+ times less space than required by the baseline.

Table E.6 shows that OC-KJL-QS and OC-Nyström-QS retain the AUC and train time of the baseline OCSVM using the STATS+HEADER features.

**Table E.6:** ***no tuning**. Retained AUC (method over OCSVM) and train time speedup (OCSVM over method) with STATS+HEADER.*

| Method \ Dataset | UNB | CTU | MAWI | MACCDC | SFRIG | AECHO | DWSHR |
|---|---|---|---|---|---|---|---|
| OC-KJL-QS: AUC Retained | 2.29 ± 0.50 | 1.02 ± 0.06 | 0.22 ± 0.23 | 0.93 ± 0.09 | 1.00 ± 0.02 | 0.52 ± 0.28 | 1.60 ± 0.09 |
| Train Speedup | 1.41 ± 0.04 | 1.24 ± 0.03 | 1.19 ± 0.04 | 1.38 ± 0.03 | 1.22 ± 0.02 | 1.28 ± 0.03 | 1.27 ± 0.03 |
| OC-Nyström-QS: AUC Retained | 2.32 ± 0.05 | 1.20 ± 0.11 | 0.13 ± 0.01 | 1.11 ± 0.06 | 0.57 ± 0.20 | 0.49 ± 0.20 | 1.70 ± 0.10 |
| Train Speedup | 1.40 ± 0.04 | 1.22 ± 0.03 | 1.31 ± 0.04 | 1.43 ± 0.03 | 1.24 ± 0.02 | 1.31 ± 0.03 | 1.24 ± 0.03 |

**Table E.7:** *OC-KJL-QS with STATS+HEADER, **no tuning**: Test time speedup (OCSVM over method) and space reduction (OCSVM over method).*

| Dataset | | UNB | CTU | MAWI | MACCDC | SFRIG | AECHO | DWSHR |
|---|---|---|---|---|---|---|---|---|
| Test Time Speedup | RSPI | 28.50 ± 0.22 | 28.91 ± 0.19 | 12.99 ± 0.07 | 28.50 ± 0.02 | 28.83 ± 0.05 | 25.42 ± 0.21 | 28.04 ± 0.11 |
| | NANO | 38.34 ± 0.39 | 36.42 ± 0.13 | 15.53 ± 0.01 | 35.37 ± 0.06 | 36.16 ± 0.08 | 29.71 ± 0.03 | 36.86 ± 0.03 |
| | Server | 38.39 ± 1.25 | 30.22 ± 0.56 | 14.20 ± 0.07 | 29.82 ± 0.30 | 30.26 ± 0.37 | 32.25 ± 0.01 | 39.59 ± 0.04 |
| Space Reduction | | 43.53 ± 0.01 | 40.89 ± 0.00 | 25.14 ± 0.00 | 40.85 ± 0.02 | 39.83 ± 0.01 | 40.65 ± 0.00 | 41.90 ± 0.01 |

**Table E.8:** *OC-Nyström-QS with STATS+HEADER, **no tuning**: Test time speedup (OCSVM over method) and space reduction (OCSVM over method).*

| Dataset | | UNB | CTU | MAWI | MACCDC | SFRIG | AECHO | DWSHR |
|---|---|---|---|---|---|---|---|---|
| Test Time Speedup | RSPI | 28.16 ± 0.22 | 28.70 ± 0.19 | 12.98 ± 0.07 | 27.15 ± 0.02 | 28.74 ± 0.05 | 24.38 ± 0.20 | 28.08 ± 0.11 |
| | NANO | 37.42 ± 0.38 | 36.29 ± 0.13 | 15.51 ± 0.01 | 33.71 ± 0.06 | 36.28 ± 0.08 | 27.86 ± 0.02 | 35.73 ± 0.03 |
| | Server | 37.60 ± 1.23 | 30.18 ± 0.55 | 14.01 ± 0.07 | 28.55 ± 0.28 | 30.39 ± 0.37 | 29.88 ± 0.01 | 37.92 ± 0.04 |
| Space Reduction | | 43.22 ± 0.01 | 40.87 ± 0.00 | 25.13 ± 0.00 | 39.94 ± 0.02 | 39.88 ± 0.01 | 39.61 ± 0.00 | 41.37 ± 0.01 |

