# OpenReview forum: "An Efficient One-Class SVM for Novelty Detection in IoT"
_TMLR — Accepted by TMLR_

### Review · Reviewer_o49b · 2022-08-17

**Summary Of Contributions:**

This paper introduces an efficient one-class SVM algorithm for anomaly detection, aiming to reduce the detection time and space. The new algorithm first leverages Nyström or KJL equation to extract embedded data for high-dimensional mapping and then fits the normal data into a Gaussian Mixture Model for anomaly detection. The experimental results show that the algorithm can speed up the detection time.

**Broader Impact Concerns:**

The paper does not raise any broader impact concerns.

**Requested Changes:**

- Explain why to use GMM in the algorithm.
- Provide a figure showing the relationship between retrained AUC and the test time speed.
- Compare the detection time with the generation time of DDoS attacks.
- Reorganize the paper.


**Strengths And Weaknesses:**

Strengths
- The paper investigates a timely problem of the efficiency of SVM in the inference phase.
- The proposed algorithm is implemented on real-world IoT devices.
- The experimental result shows a large improvement in both detection time and space.

Weaknesses
- The novelty of the paper is a bit limited. The proposed algorithm is built upon the existing Nyström or KJL approach with a well-established GMM model.
- Why does the paper fit the normal data into a GMM model? Why not use other unsupervised clustering algorithms?
- It is hard to understand the relationship between retrained AUC and the test time speed up in Table 2, 3, and 4.
- The paper evaluates the efficiency of one-class SVMs on DDoS attacks. However, since DDoS attacks generate a large amount of traffic in a short time, it is unclear whether the proposed algorithm can be efficient enough to detect the attacks in real time. It would be better if the paper could compare the speed of DDoS attacks with the speed of detection.
- The paper is not well-organized and hard to follow.

---

### Review · Reviewer_tP4r · 2022-08-19

**Summary Of Contributions:**

- The authors tried to solve the novelty detection problems in IoT domains with the constraints of low memory and low latency (at inference).
- The authors claimed that the current OC-SVM's prediction performances are satisfactory; however, its memory and latency constraints are not practical to utilize in IoT domains.
- The authors proposed an approximation of original OC-SVM and combined with GMM to significantly reduce the necessary memory size and the latency at inference time only with marginal performance loss.
- The experimental results show that the latency and memory reduction are significant (x20) which makes OC-SVM practical in IoT settings.

**Requested Changes:**

**1. Is OC-SVM state-of-the-art?**
- In order to claim that the OC-SVM is SOTA, the authors should provide some quantitative evidence.
- At least for the datasets that the authors used in this paper, the authors should provide some alternative anomaly detection methods (instead of OC-SVM) and discuss the performances of those alternatives. - Something like the accuracy and the inference time.
- Currently, it is hard to defend why we need to focus on OC-SVM instead of many other anomaly detection algorithms.

**2. Unsupervised setting**
- The authors claim that the novelty detection settings are unsupervised settings. In other words, only the normal samples are provided and the abnormal samples are not provided.
- It is unclear how to gather only normal samples. In experiments, the datasets include both normal and abnormal samples (which is critical to evaluate the anomaly detection performances). - So, how to extract only normal samples from the unlabeled data?
- Please clarify why the gathering of the anomaly samples is prohibited and how to gather only normal samples in practice.
- Also, "if" we can use the gathered anomaly samples as well for constructing the models (like supervised model training), is it better than OC-SVM?

**3. GMM and alternatives**
- The inference time latency is heavily relied on GMM.
- As the authors said, the main advantage of GMM is a fairly simple inference procedure which would significantly decrease the latency and memory.
- In that point of view, is it better to just use GMM instead of using OC-SVM first then convert it to the GMM forms?
- Note that the prediction performances of GMM are also quite promising in multiple anomaly detection applications.
- If the kernel is the problem, we can use the kernel density estimator (https://scikit-learn.org/stable/modules/density.html#kernel-density-estimation) directly.

**4. Figure 3**
- In Figure 3 (Data), even though it is not easy to separable with a linear decision boundary; however, with a simple kernel, we can separate two classes easily.
- For instance, if we directly use the GMM, we can easily solve this anomaly detection example without having an approximation of OC-SVM.
- It would be good if the authors can provide some examples that OC-SVM approximation can solve but GMM cannot easily solve in the revised manuscript.

**Strengths And Weaknesses:**

**Strength**

- The authors tackled an important and practical problem: anomaly detection with the constraints of memory and latency.
- The proposed method makes sense and the experimental results are significant.

**Weakness**

- Some of the claims do not have enough supports. For instance, the authors should provide more quantitative analyses on why OC-SVM is SOTA in this area.
- The authors did not properly consider alternatives such as directly applying GMM or KDE for anomaly detection method.
- The problem settings are unclear. Unsupervised usually means unlabeled data. But the authors used the term unsupervised as only with normal samples. As I understand, if we want to gather only with normal samples, we should exclude all the anomalies from the unlabeled data. It should be clarified in the rebuttals.

---

### Review · Reviewer_MEPD · 2022-08-23

**Summary Of Contributions:**

The paper targets an important problem to make a test-time efficient anomaly detection method in low-resource hardware (e.g., IoT). To this end, the paper target to make OCSVM (which shows an effective performance in this area) efficient. The main contribution of this paper is to extend the Nystrom and (Gaussian) Sketching approaches (used in efficient SVM) into OCSVM by combing with clustering and GMM. The experimental results show that the proposed method indeed improves the memory and detection time highly efficient.

**Broader Impact Concerns:**

No concerns

**Requested Changes:**

1) Some experiments to support that the OCSVM is the current state-of-the-art

2) Justification of using GMM (mentioned in the weakness part)

3) More baselines (e.g., KDE, IF, subsample training set for OCSVM) by comparing both efficiency (i.e., wall-clock-time) and effectiveness (i.e., AUROC)
- 2D plot will be much more readable by putting efficiency on the x-axis and effectiveness on the y-axis

4) Clearer presentation
- Detailed explanations in the Table caption, especially Table 2 (mentioned in the weakness part)
- The introduction and Section 5,6,7 (mentioned in the weakness part)

5) (minor suggestion for improvement) Considering non-IoT datasets will strengthen the paper. Since the proposed method is can be applied to any scenario, showing non-IoT datasets will be worth it, e.g., MNIST.


**Strengths And Weaknesses:**

**Strengths**

The target problem (i.e., reducing the test time and memory) is interesting and well-motivated

The writing itself is clear, but maybe reorganizing the paper will improve the overall presentation.

------------

**Weakness**

Hard to agree that the OCSVM is the state-of-the-art method.
- There exists various (non-deep learning) methods such as KDE, IF, Mahalanobis distance, and Minimum Covariance Determinant estimator (MCD).
- The detection time of Mahalanobis and MCD does not depend on the training dataset size.

While the author claim that GMM is a natural selection for unsupervised clustering, it is slightly hard to agree (since there is no theoretical explanation). Since the main algorithm is highly dependent on the GMM, proving some evidence will strengthen the paper’s claim, e.g., empirical support by comparing more clustering algorithms.

Table 2 is quite confusing. Can the author provide the actual AUC value rather than the Retained AUC..? (Does Retained AUC > 1 mean that the proposed method is better than the original OCSVM?)

According to Figure 2, reducing the training data size did not reduce the AUC much. Hence, one naive way is to randomly subsample the training data to reduce the time and memory complexity of OCSVM (a better way is to select the prototypical sample, e.g., using the k-medoids clustering). Does the proposed method perform better than this naive way?

The overall presentation can be improved.
- For instance, a more abstract version of the Introduction section will be more readable (especially from the results overview).
- Also, making Sections 5,6,7 into one section and divide with subsections will be much better.
- The Table captions are hard to understand.

[1] Parzen et al., On Estimation of a Probability Density Function and Mode, 1962\
[2] Liu et al., Isolation Forest, ICDM 2008\
[3] Mahalanobis et al., "On the generalised distance in statistics, 1936

---

### Decision · Action_Editors · 2022-10-03

**Recommendation:** Accept with minor revision

**Comment:**

The paper presents an efficient implementation of One-Class Support Vector Machines (OCSVM). The authors first apply the Nystrom and (Gaussian) Sketching approaches as used in the SVM literature, and then use GMM models for the detection. Reviewers highlighted the importance of the problem and significance of the results. However, there are several concerns needed to be addressed by the authors, e.g., (a) the authors say OCSVM is a SOTA method, but it is hard to agree, and (b) why not use GMM instead of OCSVM? AE carefully read the authors' response as well as the revised draft. Overall, AE thinks the authors provide reasonable answers (and supporting additional experimental results), and such concerns should not be a reason to reject the paper (under the TMLR's acceptance policy). Nevertheless, to be accessed by a broader audience, AE strongly recommend the authors to provide more motivations and intuitions for OCSVM and GMM. AE also finds many typos, which should be fixed before publishing the draft, e.g., (c) OCSVM has the performance in AUC than others (performance -> better performance?), and (d) we notice that GMM has a higher AUC than OCSVM over different training sizes (GMM -> KDE?).